

**BrGDGTs-based seasonal paleotemperature reconstruction for the last 15,000 years**
**from a shallow lake on the eastern Tibetan Plateau**
Xiaohuan Hou [a], Nannan Wang [a], Zhe Sun [b], Kan Yuan [a, c], Xianyong Cao [a], Juzhi Hou [a *]
[a] *Group of Alpine Paleoecology and Human Adaptation (ALPHA), State Key Laboratory of Tibetan*
*Plateau Earth System, Resources and Environment (TPESRE), Institute of Tibetan Plateau Research,*
*Chinese Academy of Sciences, Beijing 100101, China*
[b.] *Institute of Geography and Resources Science, Sichuan Normal University, Chengdu, 610066, China*
[c.] *University of Chinese Academy of Sciences, Beijing 100049, China*
* Corresponding author
E-mail address: houjz@itpcas.ac.cn



**ABSTRACT**
Knowledge of Holocene temperature changes is crucial for addressing the problem of the
discrepancy between Holocene proxy temperature reconstructions and climate model
simulations. The complex spatiotemporal pattern of temperature variations on the Tibetan
Plateau (TP) further complicates the study of Holocene continental climate change. The
discrepancy between model-based and proxy-based Holocene temperature reconstructions
possibly results from the seasonal biases and environmental ambiguities of the proxies.
Quantitative temperature reconstructions using different proxies from the same sediment core
can provide an effective means of evaluating different proxies; however, this approach is
unusual in terrestrial environments. Here, we present an ice-free-season temperature record
for the past 15 ka from a shallow, freshwater lake on the eastern TP, based on brGDGTs
(branched glycerol dialkyl glycerol tetraethers). This record shows that the Holocene Thermal
Maximum lags the pollen-based July temperature recorded in the same sediment core. We
conclude that the mismatch between the brGDGTs-based and pollen-based temperatures is
primarily the result of seasonal variations in solar irradiance. The overall pattern of
temperature changes is supported by other summer temperature records, and the Younger
Dryas cold event and the Bølling–Allerød warm period are also detected. A generally warm
period occurred during 8–3.5 ka, followed cooling in the late Holocene. Our findings have
implications for understanding the seasonal signal of brGDGTs in shallow lakes, and provide
critical data for confirming the occurrence of seasonal biases in different proxies from high-
elevation lakes. To further investigate the significance of the brGDGTs and temperature
patterns on the TP, we reviewed previously published brGDGTs-based Holocene temperature



records across the TP. The results demonstrate that brGDGTs can record both annual mean
temperature and a warm-biased temperature, and that both show a gradual warming trend
during the Holocene with relatively cooler conditions during the middle Holocene, and a
cooling trend during the middle to late Holocene. We analyzed the possible reasons for the
diverse brGDGTs records on TP and emphasize the importance of considering lake
conditions and modern investigations of brGDGTs in lacustrine systems when using
brGDGTs to reconstruct paleoenvironmental conditions.
**Keywords:** Tibetan Plateau, brGDGTs, warm-biased temperature, shallow lake, Holocene
**1 Introduction**
Global climate change has had a profound impact on both the natural ecological and socio-
economic systems that are vital for human survival and development, making climate change
a critical limiting factor for the sustainable development of human society. The Tibetan
Plateau (TP), also called the "Third Pole" (Qiu, 2008), has undergone rapid warming over the
last five decades, with a rate twice that of the global average (0.3 – 0.4°C/decade) (Chen et al.,
2015; Kuang and Jiao, 2016), making it one of the world's most temperature-sensitive regions
(Chen et al., 2015; Yao et al., 2022). Consequently, assessing the impact of future climate
change on the TP is becoming increasingly important. To enhance the precision and accuracy
of future climate change estimates for the TP under ongoing global climate change and to
minimize the uncertainty in climate simulations, it is essential to investigate the processes and
mechanisms of regional climate and environmental changes, with particular emphasis on
temperature, on a relatively long timescale, such as that of the Holocene.



The Holocene, the most recent geological epoch, is closely linked with the development of
human civilization. Quantitative reconstructions of Holocene temperature trends can be used
to explore their impacts on civilization and to establish a geological and historical context for
predicting future climate changes. In recent decades, several Holocene quantitative
reconstructions of seasonal and annual temperatures for the TP have been produced using
various proxies, like pollen (Lu et al., 2011; Herzschuh et al., 2014), chironomids (Zhang et
al., 2017; Zhang et al., 2019a), $\delta^{18}$O in ice deposits (Thompson et al., 1997; Pang et al., 2020),
and biomarkers (Zhao et al., 2013; Hou et al., 2016; Cheung et al., 2017). These
reconstructions have provided crucial data for the elucidation of Holocene temperature
changes. However, the available Holocene temperature records from the TP show divergent
trends. Multiple proxy indicators indicate three different Holocene temperature patterns on
the TP. First, a consistent Holocene warming trend (Opitz et al., 2015; Feng et al., 2022; Sun
et al., 2022). For example, brGDGTs based annual temperatures (Feng et al., 2022; Sun et al.,
2022) indicate a gradual warming trend which resembles the $\delta^{18}$O temperature record from
the Chongce ice core on the western TP, except for the last 2 ka (Pang et al., 2020). Second,
an early to middle Holocene summer temperature maximum and a gradual cooling trend
during the late Holocene are observed in pollen-, alkenone- and chironomid-based
temperature records (Herzschuh et al., 2014; Zheng et al., 2015; Hou et al., 2016; Zhang et al.,
2017; Wang et al., 2021a). Third, a prominent relatively cool middle Holocene (Li et al., 2017;
Wang et al., 2021c); for example, a composite temperature record suggests that temperatures
were ~2°C cooler during the middle Holocene than during the early and late Holocene (Wang
et al., 2021c). Several records also show a steady long-term trend without distinct cooling or



warming (Sun et al., 2021). Moreover,  the cooling trends in proxy-based Holocene
temperature records are inconsistent with those of climate models, which indicate a warming
trend, and this inconsistency is widely known as the "Holocene temperature conundrum" (Liu
et al., 2014). There are several potential factors that may contribute to the disparity in
Holocene temperature trends, including seasonal biases and uncertainties in temperature
proxies and reconstructions, independent of climate models (Liu et al., 2014; Marsicek et al.,
2018; Hou et al., 2019; Bova et al., 2021; Cartapanis et al., 2022). While several recent
studies have suggested that seasonality in proxies is not the major cause of the Holocene
temperature conundrum (Dong et al., 2022; Zhang et al., 2022b), it is significant that the TP
is an alpine and high-altitude region with significant seasonal temperature variations.
Moreover, most organisms tend to grow during the warmer seasons at high latitudes and high
altitudes (Zhao et al., 2021a). Currently, however, we lack unambiguous and reliable seasonal
temperature records to support a seasonality-bias hypothesis. Most previous studies have
relied on a single temperature proxy, and the few studies that have used multiple proxies from
the same sediment core have tended to focus on annual average temperature and summer
temperature. For example, a chironomid-based July temperature reconstruction for Tiancai
lake on the southeastern TP shows higher temperatures during the early to middle Holocene
(Zhang et al., 2017), while the brGDGTs-based annual average temperature shows a warming
trend (Feng et al., 2022). Different proxies may reflect the seasonal temperatures in different
months, and thus producing temperature reconstructions for different months for the same
sediment core may help better understand the seasonal bias of terrestrial temperature records.
Furthermore, the reconciliation of the divergent trends of Holocene temperature on the TP



and its surroundings requires additional high-altitude temperature records from these regions,
with reliable chronologies and proxy records with an unambiguous climatological
significance.

Branched glycerol dialkyl glycerol tetraethers (brGDGTs) are a group of membrane-spanning
lipids found in bacteria (Fig. S1) (Damsté et al., 2000; Chen et al., 2022; Halamka et al.,
2022), and they have become a powerful tool for quantifying past terrestrial temperature
variations. Through investigations of brGDGTs in globally-distributed soils, it was found that
the  distribution of brGDGTs is primarily related to temperature and pH (Weijers et al., 2007).
Subsequently, brGDGTs–temperature calibrations from soil, peat and lake sediments were
established on scales from global (Weijers et al., 2007; De Jonge et al., 2014; Crampton-
Flood et al., 2020; Martínez-Sosa et al., 2021) to regional (e.g., East Asia) (Sun et al., 2011;
Ding et al., 2015; Wang et al., 2016; Dang et al., 2018), leading to significant progress in
reconstructing terrestrial temperatures, particularly on the TP (Zhao et al., 2013; Cheung et
al., 2017; Li et al., 2017; Zhang et al., 2022a).

Natural lakes are widely distributed across the TP (Zhang et al., 2019b). Lake sediments are
often organic matter-rich and they accumulate continuously and rapidly, providing high
resolution records of environmental change, and they are thus regarded as the most important
terrestrial climate archive (Moser et al., 2019). BrGDGTs in lacustrine systems are often
more strongly correlated with temperature, with higher coefficient of determination ($r^2$) and
lower root mean square error (RMSE) values (Martínez-Sosa et al., 2021), than in soils and



peats. However, the factors influencing the distribution of brGDGTs in lakes are complex and
multidimensional; moreover, as well as temperature and pH, other factors like salinity (Wang
et al., 2021b), oxygen content (Buckles et al., 2014a), and water depth (Woltering et al., 2012)
can significantly impact the distribution of brGDGTs in lakes.

In this study, we obtained a quantitative temperature reconstruction for the past 15 ka from
Gahai, a shallow (average depth of ~2 m) freshwater lake located in the source area of the
Yellow River. This region is an important ecological protection area on the eastern edge of
the TP. Freshwater environments avoid the confounding effects of salinity on brGDGTs-
based temperature reconstructions, and shallow lakes also minimize the impact of the uneven
distribution of light and nutrients on brGDGTs. Our specific aims were: (1) to determine the
long-term trend of Holocene warm-biased terrestrial temperatures at a high elevation; (2) to
compare records of ice-free season temperatures with July temperatures from the same
sediment core; and (3) to gain a better understanding of the possible mechanisms responsible
for Holocene temperature variations, especially on the TP.
**2 Materials and methods**
*2.1 Study site*
Gahai (102°11′–102°28′ E, 34°04′–34°4′ N, 3444 m a.s.l.) is a freshwater lake and part of the
Gahai meadow wetland, which is a national nature reserve with restricted human access, on
the eastern edge of the Tibetan Plateau (Fig. 1). The lake is fed by runoff from the
surrounding hills, and it drains into the Tao river, which ultimately enters the Yellow river.
Thus, Gahai lake is a critical water conservation area in the upper reaches of the Yellow River.



The average water depth of Gahai is ~1–2 m, and the maximum depth is ~5 m. The
vegetation in the catchment consists mainly of *Kobresia tibetica, Equisetum arvense,*
*Potentilla anserina, Artemisia subulate,* and *Oxytropis falcata* (Ma et al., 2019).
Meteorological data for the area are available from Langmu Temple station (Fig. 1) (102°38′
E, 34°5′ N, 3412 m a.s.l.), ~32 km northwest of Gahai lake. They indicate an annual average
(mean) precipitation of 781 mm, with > 67% occurring between June and September, and
mean annual temperature of 1.2 °C with a relative humidity of ~65%. The summers are mild
and humid and the winters are cold and dry. From May to September, the mean average
temperature is above freezing (0°C), but the temperature in May is very low, close to 0°C.

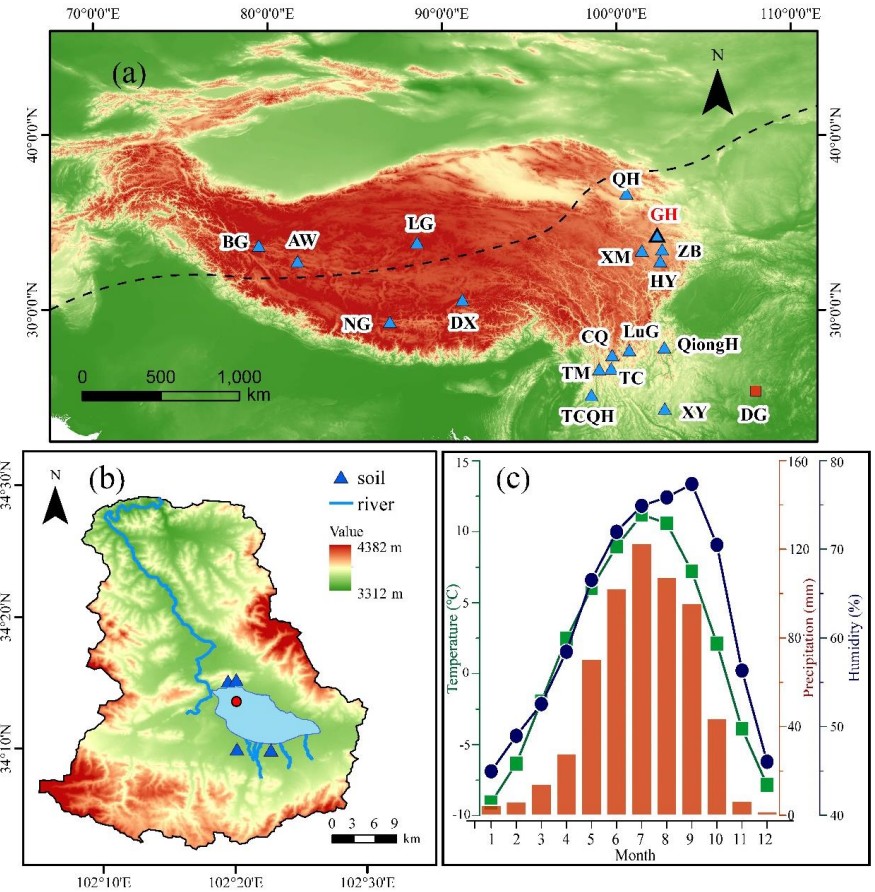


**Fig. 1** (a) Locations of the sites on the Tibetan Plateau referenced in the text. Triangle with

bold line indicates the location of Gahai lake (this study). Other triangles indicate the

locations of cited studies on the Tibetan Plateau and the surrounding area: Bangong Co

(BG), Aweng Co (AW), Ngamring Co (NG), Linggo Co (LG), Dangxiong wetland (DX),

Qinghai lake (QH), Ximen Co (XM), Zoige Basin (ZB), Hongyuan peatland (HY), Lugu

lake (LuG), Cuoqia lake (CQ), Tingming lake (TM), Tengchongqinghai lake (TCQH),

Tiancai lake (TC), Qionghai lake (QH), Xingyun lake (XY). Red square indicates

Dongge Cave (DG). Black dotted line represents the northern boundary of the modern

Asian summer Monsoon (Chen et al., 2008). (b) Drainage basin of Gahai lake and the

core site. (c) Climate data from Langmu Temple meteorological station: monthly

temperature (green line), precipitation (red bars), and humidity (blue line).





*2.2 Sampling*
A sediment core with the length of 329 cm was obtained from Gahai Lake in January 2019, at
a water depth of 1.95 m, using a UWITEC platform operated from the frozen lake surface. In
addition, several catchment soil samples were collected from around the lake (Fig. 1). All
samples were transported to the Institute of Tibetan Plateau Research, Chinese Academy of
Sciences (ITPCAS). The sediment core was split lengthwise, and one half was subsampled
and freeze-dried for subsequent analysis.

*2.3 Chronology*
The chronology of the upper 20 cm of the sediment core is based on measurements of [210]Pb
and [137]Cs, at a 1-cm interval. The chronology for the deeper part of the core is provided by
accelerator mass spectrometry (AMS) [14]C measurements of 13 bulk sediment samples, which
were conducted by Beta Analytic Inc. (Miami, USA) (Fig. 2) (Wang et al., 2022).

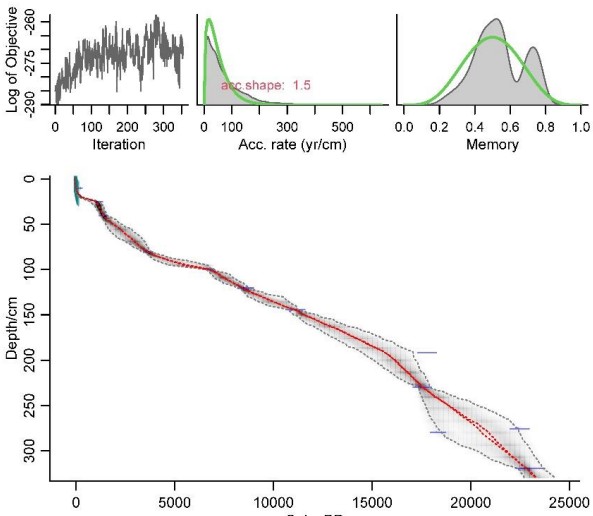


**Fig. 2** Age-depth model for Gahai, based on AMS [14]C, [210]Pb and [137]Cs ages (Wang et al.,
2022). The ages of the upper 20 cm are based on [210]Pb and [137]Cs dating (green symbols)




and those of the lower part on AMS [14]C dates (blue symbols).

*2.4 Lipids extraction and brGDGTs analysis*
For lipids extraction, ~5 g samples were ground to a powder and extracted ultrasonically with
dichloromethane (DCM): methanol (MeOH) (9: 1, v: v) three times. The supernatants were
combined and dried under a stream of nitrogen gas. Subsequently, the total lipid extracts were
separated into neutral and acid fractions through a LC-NH$_2$ silica gel column using DCM:
isopropyl alcohol (2: 1, v: v) and ether with 4% acetic acid (v: v), respectively. The neutral
fraction was then eluted through a silica gel column using n-Hexane, DCM and MeOH, and
the GDGTs were dissolved in the MeOH. The GDGTs fraction was passed through a 0.45 μm
polytetrafluoroethylene (PTFE) filter before analysis. C$_{46}$-GDGT (a standard compound)
(Huguet et al., 2006) was added to the samples before analysis.

BrGDGTs were detected using an HPLC-APCI-MS (Waters ACQUITY UPLC I-Class/Xevo
TQD) with auto-injection at the ITPCAS. The compounds were separated by three Hypersil
Gold Silica LC columns in sequence (each 100 mm × 2.1 mm, 1.9 μm, Thermo Fisher
Scientific; USA), maintained at a temperature of 40℃. GDGTs were eluted isocratically
using 84% hexane and 16% ethyl acetate (EtOA) for the first 5 min, followed by a linear
gradient change to 82% hexane and 18% EtOA from 5 to 65 min. The columns were cleaned
using 100% EtOA for 10 min, and then back to 84% hexane and 16% EtOA to equilibrate the
column, with a flow rate of 0.2 ml min$^{-1}$.



The APCI-MS conditions were as follows: nebulizer pressure at 60 psi, APCI probe
temperature at 400℃, drying gas flow rate of 6 L/min and temperature of 200℃, capillary
voltage of 3600 V, source corona of 5.5 µA. Detection was performed in selected ion
monitoring (SIM) mode, targeting the protonated molecules at m/z 1050, 1048, 1046, 1036,
1034, 1032, 1022, 1020, 1018 and 744. The results were analyzed using MassLynx V4.1
software, and quantification was achieved by comparing the peak areas of targeted ions and
the internal standard, assuming an identical response factor for GDGTs.

**3 Results and Discussion**

*3.1. Concentration and distribution of brGDGTs in the sediment core and catchment soils*
BrGDGTs were detected in both the catchment soils and the downcore sediments. The
average concentration of brGDGTs in the catchment soils (0.07 ng g$^{-1}$dw) was significantly
higher than in the surficial core sediments (0.70 ng g$^{-1}$dw). In the soil samples,
pentamethylated brGDGTs were generally the most abundant (55.33%), followed by
tetramethylated brGDGTs (23.60%) and hexamethylated brGDGTs (21.07%) (Fig. S2). The
relative amount of cyclopentane ring-containing brGDGTs in the soil samples was generally
low (24.34%) and it was sometimes too low to be detected, especially the fractions of IIIb,
IIIb', IIIc, IIIc', IIc and IIc'. In the downcore sediments, the relative abundance of
tetramethylated brGDGTs (43.84%) was like that of pentamethylated brGDGTs (41.93%),
and hexamethylated brGDGTs were the least abundant (14.22%) (Fig. S2). The relative
abundant of cyclopentane ring-containing brGDGTs in the downcore sediments (67.82%)
was lower than that in the catchment soils.





*3.2 In situ production of brGDGTs in Gahai lake*
Although lacustrine brGDGTs have great potential for quantitatively reconstructing terrestrial
paleotemperatures, uncertainties about their sources in lacustrine environments are a major
factor limiting their application (Damsté et al., 2009; Tierney and Russell, 2009; Sun et al.,
2011; Buckles et al., 2014b; Cao et al., 2020). To investigate the origin and characteristics of
brGDGTs in the Gahai lake sediments, we examined the distributions and concentrations of
brGDGTs in the sediments and catchment soils and found significant differences between
them. First, as described in the previous section, the average content of brGDGTs in the
catchment soils was ~10% that of the surficial lake sediments, suggesting the absence of
large-scale allochthonous inputs from the catchment soils. Second, the brGDGTs distributions
in the downcore sediments were quite different from those in the catchment soils, which
suggests a significant autochthonous brGDGTs contribution to the lake sediments (Fig. 3 and
Fig. S2). Moreover, the ratios of 6-methyl brGDGTs to 5-methyl GDGTs ($IR_{6ME}$) in the soils
and sediments, calculated according to the formula proposed by De Jonge et al. (2014), were
significantly different. In the soil samples, $IR_{6ME}$ varied between 0.54 and 0.57 and the
average ratio in the downcore samples was 0.26, varying between 0.18 and 0.47. Third, the
in-situ production of brGDGTs in Gahai lake is suggested by the discrepancies in the degree
of methylation ($MBT'_{5ME}$) between the soils and surface sediments. The average value of
$MBT'_{5ME}$ in the Gahai lake surface sediments was 0.48, which is clearly higher than in the
catchment soils, with the range of 0.32–0.35. Fourth, and potentially the most significant, the
IIIb'and Ib' compounds are present in the catchments soil but not in the Gahai lake surficial
sediments, which may be direct evidence of an autochthonous brGDGTs contribution in the





lacustrine environment (Fig. 3), and a lower proportion of soil-derived brGDGTs input.
Therefore, we conclude that the brGDGTs in the Gahai lake sediments are mainly of in-situ
origin.

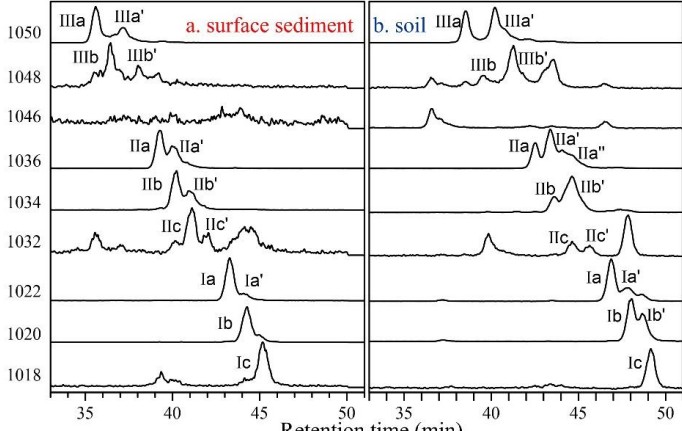


**Fig. 3** Representative high-performance liquid chromatography/atmospheric pressure
chemical ionization-mass spectrometry (HPLC/APCIMS) chromatograms of brGDGTs
from (a) surface sediments from Gahai lake, and (b) soils in the catchment of Gahai lake.

*3.3 brGDGTs-temperature calibration and Holocene temperature reconstruction*
Given the substantial contribution of authigenic brGDGTs in the Gahai lake sediments, we
reconstructed the Holocene paleotemperature record using previously published lake-specific
brGDGTs-temperature calibrations (e.g., Sun et al.,2011; Günther et al., 2014; Wang et al.,
2016; Dang et al., 2018; Russell et al., 2018; Martínez-Sosa et al., 2021). As shown in Fig. S3,
most calibrations produced qualitatively similar patterns of temperature change when applied
to the sediment core from Gahai lake, but the amplitudes vary considerably. Among these
calibrations, the reconstruction based on Martínez-Sosa et al. (2021) was chosen to produce





the final result, for several reasons. We compared the fractional abundances of summed tetra-,
penta- and hexamethylated brGDGTs of Gahai lake with other datasets (Fig. 4), including
lake sediments from the Tibetan Plateau (Günther et al., 2014; Wang et al., 2016), East Africa
(Russell et al., 2018), and global lakes (Martínez-Sosa et al., 2021). The fraction plot of the
Gahai core sediments is clearly distinct from the other Tibetan Plateau lake-sediments, even
though they are all from the same region (Fig. 4), likely because the brGDGTs in Tibetan
lakes are mainly soil-derived (Wang et al. (2016). Moreover, the novel analytical technique
for separating 5- and 6-methyl isomers was not used in the studies of Wang et al. (2016) and
Günther et al. (2014), and thus these two calibrations were excluded. The fractional
distribution of brGDGTs in Gahai lake is spanned by that of global lakes, and based on
multiyear observed temperature records from the nearest meteorological station, the modern
mean temperature of the months with temperatures above freezing in Gahai lake (May to
September) was 8.8°C, which is like the brGDGT-inferred temperature for the surficial
sediments (9.4°C), obtained using the calibration of Martínez-Sosa et al. (2021). However,
the annual mean temperature reconstructed according to Russell et al. (2018) differs
significantly from that from Langmu Temple station, although the characteristics of the Gahai
brGDGTs fractions resemble those of East African lakes. The paleotemperature
reconstruction for Gahai lake based on the warm season-temperature calibration proposed by
Dang et al. (2018) is similar to that of Martínez-Sosa et al. (2021); however, this calibration
was established based on an investigation of 35 Chinese alkaline lakes, in contrast to
freshwater Gahai lake. Similarly, although the salinity effect was corrected, the calibration
reported by Wang et al. (2021b) is not considered here. Therefore, we used a new Bayesian



calibration for the mean temperature of the Months Above Freezing (Martínez-Sosa et al.,
2021) to reconstruct a warm-biased temperature for Gahai lake.

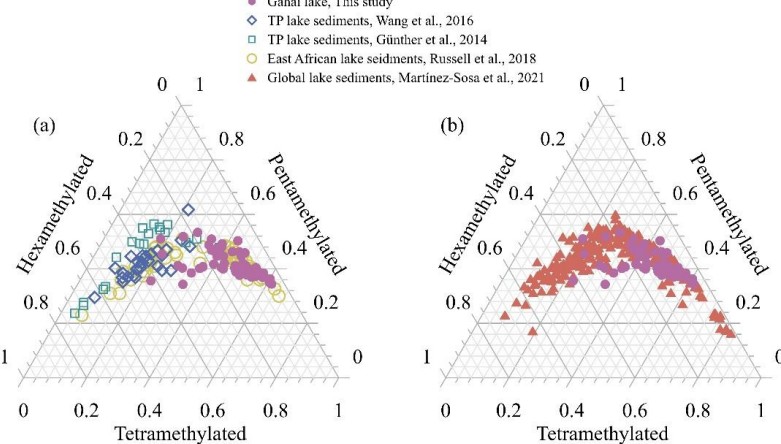


**Fig. 4** Comparison of the fractional abundances of tetramethylated, pentamethylated, and

hexamethylated bGDGTs in sediment core samples from Gahai with lake surface

sediments from the Tibetan Plateau (Günther et al., 2014; Wang et al., 2016), East Africa

(Russell et al., 2018), and worldwide (Martínez-Sosa et al., 2021).


Many studies have suggested that lacustrine brGDGTs-derived temperatures are likely to
have a warm season bias, especially in cold regions at middle to high latitudes (Shanahan et
al., 2013; Peterse et al., 2014; Dang et al., 2018; Cao et al., 2020). However, for lakes in
warmer regions, the reconstructed temperatures are much closer to the annual average
temperature (Tierney et al., 2010; Loomis et al., 2012). Gahai is a shallow lake that is usually
completely frozen during winter and spring, and the local meteorological data show that the
average snowfall period is 269 days, and that the snowfall period lasts for ~50 days. Thus, the



light transmittance and oxygen content during the lake water freezing season at Gahai are
reduced, as well as the lake water nutrient contents, which seriously inhibit the growth of
autotrophic microorganisms. Although the bacteria that produce brGDGTs are not well
characterized, heterotrophic bacteria will be reduced by the decreased autotrophic biomass.
Therefore, we suggest that the brGDGTs-based temperatures from Gahai are biased towards
the growing season (summer and autumn).

The depth interval of 191–279 cm in the Gahai sediment core represents an interval of rapid
allocthonous sedimentation, or alternatively a slump, and therefore the results for the
corresponding time interval of 20–15 ka may be unreliable. Thus, our warm-biased
temperature record from the eastern TP spans the past 15 ka, with the average temperature of
4°C, as shown in Fig. 5a. Weak warming occurred during 14.8–11.8 ka which coincides with
the Bølling–Allerød (B/A) interstadial, and a minor cold reversal occurred during 11.8–10.5
ka, which approximates the Younger Dryas (YD). The temperature record indicates a colder
period during 11.5–8.0 ka. During 8.0–3.5 ka, Gahai experienced a stable warm period with
the average temperature of ~16.5°C, after which the temperature decreased gradually. Overall,
the maximum temperature difference since 15 ka was ~10°C. The interval of 11.5–10.5 ka is
represented by a relatively low number of samples because the concentration of brGDGTs
was below the detection limit.

*3.4 Holocene temperature changes on the eastern edge of TP and their origin*
Despite the difference in amplitude, the warm-biased temperature record from Gahai



resembles the pollen record and the pollen-based temperature reconstruction from the same
site (Fig. 5) (Wang et al., 2022). However, the brGDGTs-based Holocene Thermal Maximum
(HTM) lags the pollen-based reconstruction (Fig. 5a, b). Wang et al. (2022) used a weighted-
averaging partial least regression approach to produce a temperature record for Gahai, based
on a modern pollen dataset (n=731) from the eastern TP. Assessment of the statistical
significance of the  pollen-based climate variables for Gahai suggests that the mean July
temperature is the most important environmental factor influencing the fossil pollen
assemblages. The brGDGTs in Gahai are indicative of summer and autumn temperatures, and
the mismatch between the temperature records inferred from brGDGTs and the pollen record
may be attributed to the difference between the solar irradiance during June–October and that
during July. Additionally, significant vegetation changes occurred in the Gahai area during
4.0–3.5 ka, when the dominant high-elevation montane forest was rapidly replaced by alpine
steppe. The poor vegetation coverage and lower soil moisture level during this period (Fig. 5c,
d) (Wang et al., 2022) would have resulted in more efficient heat absorption, causing surface
warming (Lu et al., 2019).





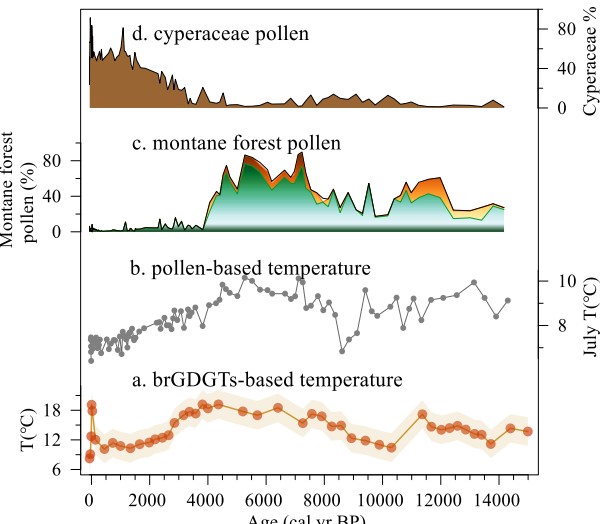

**Fig. 5** Comparison of multiproxy records from Gahai lake. (a) brGDGTs-based warm-bias

temperature (this study). (b) Temperature of the warmest month (July) based on pollen

assemblages (Wang et al., 2022). (c, d) Pollen-reconstructed montane forest (*Pinus*,

*Picea*, *Abies*) and Cyperaceae pollen record (Wang et al., 2022).

The brGDGTs-based temperature record from Gahai is also consistent with several other

pollen and pollen-reconstructed temperature records from the eastern TP (Fig. 6), suggesting

that it is a reliable representation of Holocene temperature changes in this region. For

example, pollen-based temperature reconstructions from Xingyun lake and Ximen Co on the

eastern TP show a early to middle HTM (9–4 ka) and a cooling trend thereafter (Fig. 6c, e)

(Herzschuh et al., 2014; Wu et al., 2018; Wang et al., 2021a). Additionally, lake water

temperature reconstructions based on subfossil chironomids from Tiancai lake (Fig. 6f)

(Zhang et al., 2017; Zhang et al., 2019a) and alkenones from Qinghai lake (Fig. 6g) (Hou et

al., 2016) show the same trends during the past 15 ka, as also shown by other pollen-based



temperature records from the TP (Chen et al., 2020). Pollen, chironomids and alkenones
mainly respond to the growing season temperatures in middle and high latitudes, and thus the
reconstructed temperature records are consistent with the variations in summer solar
irradiance. Similar variations were documented in temperature reconstructions at a global
scale (Marcott et al., 2013; Cartapanis et al., 2022).

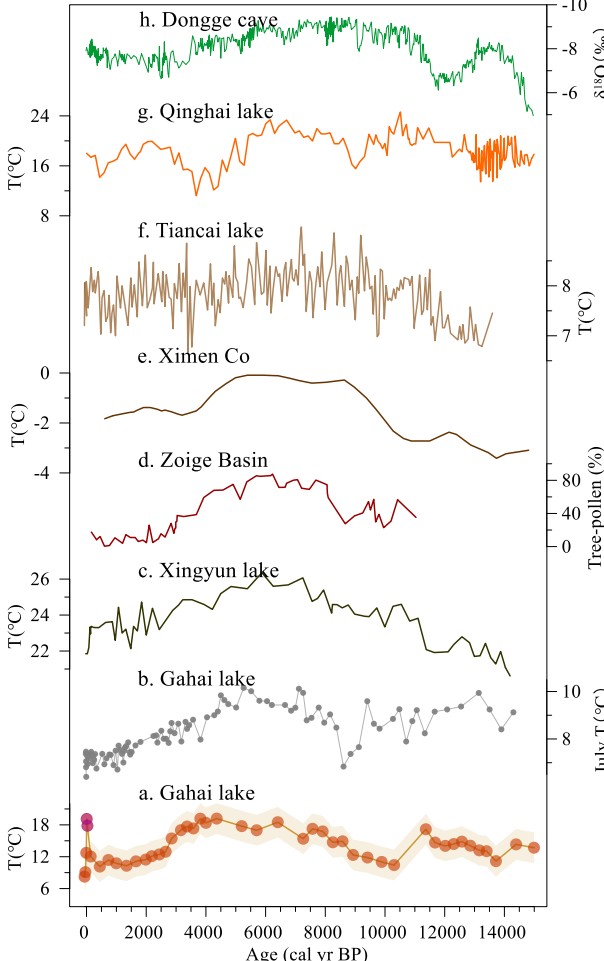


**Fig. 6** Comparison of temperature at Gahai and other records from the eastern edge of the

Tibetan Plateau. (a) brGDGTs-based warm-bias temperature at Gahai, the purple dots

may indicate unreliable temperature changes influenced by human activities (this study).





(b) Temperature of the warmest month (July) based on pollen data from Gahai (Wang et
al., 2022). (c) Pollen-based temperature at Xingyun lake (Wu et al., 2018). (d) Tree
pollen percentages from the Hongyuan peatland in the southern Zoige Basin (Zhou et al.,
2010). (e) Pollen-based temperature at Ximen Co (Herzschuh et al., 2014). (f)
Chironomid-based temperature at Tiancai lake (Zhang et al., 2017, 2019a). (g)
Alkenone-based temperature at Qinghai lake (Hou et al., 2016). (h) Stalagmite δ¹⁸O
record of Donge cave (Dykoski et al., 2005).

Nevertheless, the timing and amplitude of the Gahai temperature fluctuations differ from
those of other temperature records from this region (Fig. 6). These discrepancies may be the
result of the chronological uncertainties of these records, and to differences in the seasonal
and spatial responses to climate forcing and feedbacks. The temperature records shown in Fig.
6 mostly refer to summer temperatures, which are primarily influenced by summer insolation.

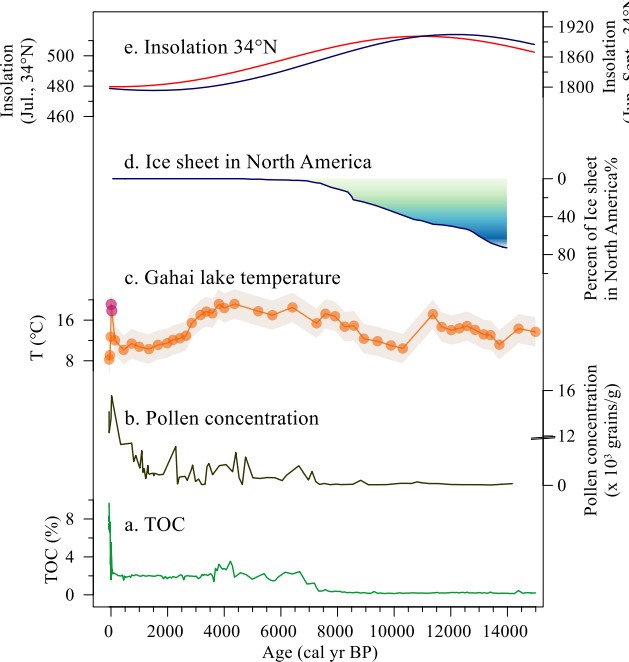






**Fig. 7** Temperature fluctuations and forcing factors during the Holocene. (a, b) TOC content

and pollen concentrations from Gahai (Wang et al., 2022). (c) brGDGTs-based warm-

bias temperature from Gahai, the purple dots may indicate unreliable temperature

changes influenced by human activities (this study). (d) Percentage of the remnant

Laurentide ice sheet in North America relative to the Last Glacial Maximum (Dyke,

2004). (e) Local insolation at 34 °N during ice-free months (Laskar et al., 2004).


The warm-biased temperature record in Gahai during the early Holocene fails to closely track
the Northern Hemisphere insolation trend, and there is also a time lag. The pollen-based
temperature record for Xingyun Lake in southwestern China also shows lower temperatures
in the early Holocene (Fig. 6c). The albedo effect caused by the increased cloud cover may be
the reason for the early Holocene decrease in summer temperatures (Wu et al., 2018).
However, the pollen record from Gahai indicates dry conditions during the early Holocene
(Wang et al., 2022), and cloud cover may not be the primary factor responsible for the low
temperatures at this time. The melting of Northern Hemisphere ice sheets during the early
Holocene weakened the Atlantic Meridional Overturning Circulation (AMOC) and
potentially also the global thermohaline circulation. This led to a reduction in the amount of
heat transport by the North Atlantic warm current to high-latitude regions, which resulted in
the low temperatures in middle to high latitudes of the Northern Hemisphere. The persistence
of the Laurentide ice sheet into the early Holocene maintained the regional albedo, as well as
discharging  meltwater into the North Atlantic (Fig. 7d) (Dyke, 2004). It is possible that these
factors impacted the summer temperatures in the Indian Summer Monsoon (ISM) domain via



ocean-atmosphere interactions. In addition, a Holocene temperature simulation showed that
global warming was more pronounced when dust factors were excluded from the simulation
(Liu et al. (2018). The record of insoluble particles in the Greenland GISP2 ice core indicates
relatively high concentrations of atmospheric aerosols in the early Holocene (Zielinski and
Mershon, 1997), which would gave weakened summer solar irradiation via radiative
feedback, leading to the cool temperatures during this period. These factors may together
have caused the early Holocene temperature decline at Gahai Lake, which slightly delayed
the onset of the Holocene Warm Period.

A significant and rapid temperature increase is evident at Gahai in recent decades, which
differs significantly from the other records (Fig. 7c). Moreover, there are notable increases in
pollen concentration, TOC, and TN (Fig. 7a, b) in the Gahai sediment core, indicating
intensive local human activities like grazing and tourism, which may be the primary cause of
the environmental changes in this region (Wang et al., 2022). This intensive human activity
may have reduced the ability of the brGDGTs to record the natural temperature background.
However, a series of environmental protection measures, including the government-enforced
exclusion of grazing, and a grassland restoration program, have been implemented to restore
the natural ecological environment of this area. Consequently, the brGDGTs-based
temperature record decreased rapidly within the modern era, returning to normal levels, and it
may provide a reliable regional record of the warm season temperature. These observations
emphasize the significant impact of human activities on climate proxies and the need to
carefully consider their effect on temperature reconstructions.




*3.5 Spatiotemporal pattern of brGDGTs-based TP temperatures*
In addition to comparing the Gahai temperature with the summer temperature records from
the eastern TP and its surrounding areas, we compiled and reviewed published Holocene
brGDGTs-based quantitative temperature records from across the TP. As shown in Fig. 8,
with the increasing number of these records for the TP, the differences between the results
have become more pronounced. The brGDGTs records from lakes in the central and western
parts of the plateau show higher temperatures in the early and late Holocene, and lower
temperatures in the middle Holocene (Li et al., 2017; He et al., 2020; Wang et al., 2021c),
while the brGDGTs records from lakes in the southern and south-eastern parts of the TP show
a warming trend throughout the Holocene (Feng et al., 2022; Sun et al., 2022). In addition,
brGDGTs in Cuoqia lake and Tingming lake, on the south-eastern TP, recorded the ice-free
season temperature, which was relatively stable during the Holocene (Sun et al., 2021; Zhang
et al., 2022a). However, our temperature record from Gahai is different from the above
records and resembles summer temperature changes during the Holocene (Chen et al., 2020).
This is because the brGDGTs record from Lake Gahai represents warm season temperatures,
which adds to its reliability.

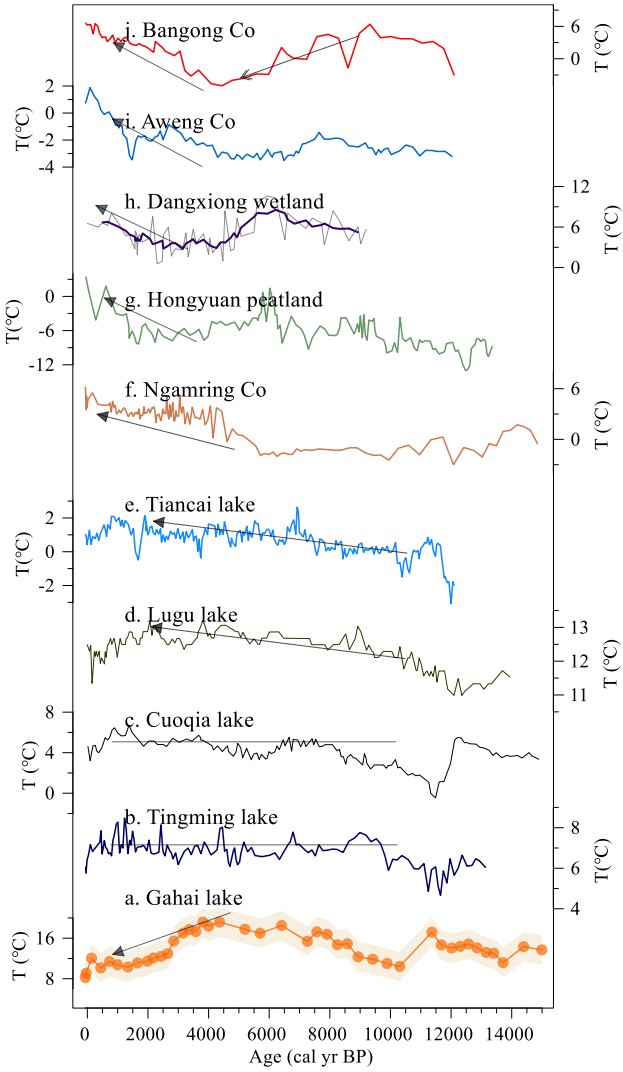

**Fig. 8** Comparison of Holocene temperature based on brGDGTs at Gahai (a) and other

records from around the TP. Reconstructed ice-free-season temperatures from (b)

Tingming lake (Sun et al., 2021), (c) Cuoqia lake (Zhang et al., 2022a). Reconstructed

annual temperature from (d) Lugu lake (Zhao et al., 2021b), (e) Tiancai lake (Feng et al.,

2022), (f) Ngamring Co(Sun et al., 2022), (g) Hongyuan peatland (Yan et al., 2021). (h)

Dangxiong wetland (Cheung et al., 2017), (i) Aweng Co (Li et al., 2017), (j) Bangong

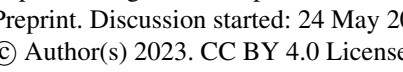


Co (Wang et al., 2021c).

We suggest that the complexity of Holocene temperature patterns recorded by brGDGTs in

TP lakes is primarily due to the ambiguity of brGDGTs in these lakes, as well as to the spatial

heterogeneity of climate change across the TP. This ambiguity can be attributed to several

factors. First, the origin of brGDGTs in lakes remains an uncertain factor in temperature

reconstruction. An increasing number of studies indicate the occurrence of a significant

amount of autochthonous brGDGTs in lakes, but their abundance in soil can also affect the

distribution of brGDGTs in lakes due to their supply via soil erosion (e.g., Tierney and

Russell, 2009; Weber et al., 2015; Wang et al., 2023). In fact, even within the same lake (e.g.,

Tengchongqinghai lake in southwestern China), two studies reached inconsistent conclusions

regarding the origin of brGDGTs (Tian et al., 2019; Zhao et al., 2021b), possibly because the

niches of certain brGDGTs may expand or contract compared to other locations within a lake.

Therefore, it is important to conduct detailed modern process studies to accurately assess the

sources of brGDGTs in lakes, especially with regard to evaluating the proportion of

autochthonous brGDGTs (Martin et al., 2020; Wang et al., 2023). Second, brGDGTs may

show a seasonal signal. Current brGDGTs–temperature calibrations for lakes reflect the

annual average temperature (Sun et al., 2011; De Jonge et al., 2014), as well as the growing

season temperature (Sun et al., 2011; Dang et al., 2018) and the ice-free season temperature

(Martínez-Sosa et al., 2021; Zhang et al., 2022a). Thus, there is no consensus regarding

whether the brGDGTs have a seasonal bias, and it is necessary to conduct continuous, high-

resolution seasonal investigations of lakes on the Tibetan Plateau to comprehensively





elucidate the seasonal characteristics of brGDGTs. This can enhance the accuracy of regional
temperature reconstruction and may help reconcile the complex temperature patterns
observed on the Tibetan Plateau. Third, the factors affecting the distribution of brGDGTs in
lakes are complex, including not only temperature, pH and salinity but also oxygen content,
water depth, and so on (Wang et al., 2016; Wang et al., 2021b). The distribution of brGDGTs
in lakes is significantly influenced by the hydrological and physical properties of the lakes,
and thus it is necessary to attain a more comprehensive understanding of the characteristics of
the lakes in the study area and their effects on brGDGTs. Fourth, different brGDGTs–
temperature calibrations may lead to significant differences in both the amplitude and trend of
temperature from the same dataset (Wang et al., 2016; Feng et al., 2019). One reason for this
is the deviation between in-situ measured temperature and atmospheric temperature (Wang et
al., 2020). Thus, selecting an appropriate calibration and attempting to establish a brGDGTs-
in situ temperature calibration are effective means of enhancing the reliability of brGDGTs-
based temperature reconstructions.

**4 Conclusions**
We present a quantitative, brGDGTs-based seasonal paleotemperature record over the last 15
ka from the sediments of a shallow lake on the eastern Tibetan Plateau. Our reconstruction
resembles the summer temperature trend, with the Holocene Thermal Maximum occurring
during 8–3.5 ka. There is a lag between our brGDGTs-based reconstruction and pollen-based
July temperature recorded in the same sediment core, indicating a significant seasonal bias
between different proxies. Since 3.5 ka, the temperature decreased gradually, and the surficial



sediments reliably recorded the warm season temperature during the current period in the
Gahai Lake region. However, intensive local human activity during the last century has
affected the distribution of brGDGTs, resulting in temperature deviations recorded by
brGDGTs. However, the implementation of environmental protection policies have reduced
this anthropogenic signal. Our findings help better understand the seasonal signal of
brGDGTs in shallow lakes and provide important data for improving projections of terrestrial
climate change at high elevations.

We also investigated previously published brGDGTs-based Holocene temperature records on
the TP to determine the pattern of brGDGTs-based temperature changes and the possible
causes of the differences between reconstructions. We emphasize the need for the careful
examination of both the source and behavior of these compounds in lacustrine environments
and lake status, prior to the application of brGDGTs proxies in paleolimnological
reconstruction.

**Competing interests**
The contact author has declared that none of the authors has any competing interests.

**Acknowledgements**
This work was financially supported by the National Natural Science Foundation of China
(41877459) and the Second Tibetan Plateau Scientific Expedition and Research
(2019QZKK0601). We would like to thank Jan Bloemendal for the help with language



editing.



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
