# Peer review of "BrGDGTs-based seasonal paleotemperature reconstruction for the last 15,000 years"

_Climate of the Past, 2023_

## Author Comment (AC1)

Dear Editor and Reviewer#1:

On behalf of the co-authors, we are very grateful to you for giving us an opportunity to revise our manuscript. We really appreciate your positive and constructive comments together with suggestions on our manuscript entitled 'BrGDGTs-based seasonal paleotemperature reconstruction for the last 15,000 years from a shallow lake on the eastern Tibetan Plateau' (MS No.: cp-2023-32). We have therefore studied reviewer' comments carefully and tried our best to revise our manuscript accordingly. Notably, the changes are highlighted in red in the revised manuscript. Please see below for a point-by-point response to the reviewers' comments and concerns.

**Responds to the comment of Reviewer#1:**

The potential seasonal bias produced by terrestrial archives are important to better understand the so-called "Holocene Temperature Conundrum", the difference between simulated global Holocene warming and proxy-reconstructed global Holocene cooling. In this manuscript, Hou et al analyze down-core brGDGTs from a sediment core collected from Gahai Lake on the eastern Tibetan Plateau. Based on brGDGTs, they reconstruct an ice-free season (May to September) temperature over the past 15 ka. They also compare the new record with a previously published pollen-based July temperature record from the same core, and find that Holocene Thermal Maximum in the ice-free season record lags that in the July temperature record. They also review other published brGDGT-based temperature records, and evaluate differences among them. They emphasize the importance of considering lake conditions and modern-process investigations when using brGDGTs to reconstruct past climate changes.

Overall, I think this study provide some valuable data to better understanding the seasonal bias in Holocene temperature reconstruction. In particular, difference records from the same sediment core from the same lake add the credibility of the results. Moreover, the systematic modern process analyzes for brGDGTs sources in lake catchment basin significantly improve the quality of the reconstructed record. Therefore, I would recommend a minor revision.

One main issue the authors should be considered is the chronology. They author simply cited the age-model results from a published paper, without any detailed explanation. For my understanding, as an independent paper, the author should necessarily and concisely explain how they reconstructed the age model and how they evaluate the potential "old carbon" effect. Although these potential age uncertainties won't affect the main finding for Holocene climate changes, they seem do affect the timing of the deglacial BA and YD events.

Response: We are very grateful to you for your meaningful comments. As you said, as an independent article, it should indeed have a detailed introduction to the chronology. We have included this part in the text, please see line 187-213.

"The chronology of the upper 20 cm of the sediment core is based on measurements of $^{210}$Pb and $^{137}$Cs, at a 1-cm interval. The chronology for the deeper part of the core is provided by accelerator mass spectrometry (AMS) $^{14}$C measurements of 13 bulk sediment samples, which were conducted by Beta Analytic Inc. (Miami, USA) (Fig. 2) (Wang et al., 2022).

The $^{210}$Pb age model was constructed using the constant rate of supply (CRS) model and the $^{137}$Cs peak was used as supplement (Appleby, 2002). The calculated age of $^{210}$Pb using CRS model aligned well with the $^{137}$Cs peak at 6 cm. Overall, the CRS model was deemed suitable for determining the age of Gahai lake.

Reservoir age, as highlighted by Hou et al. (2012), is a crucial factor affecting the age determination of lake sediment cores on the TP. Therefore, it was necessary to establish the reservoir age of Gahai lake before undertaking paleoclimate reconstruction. The linear extrapolation relationship between the $^{14}$C ages and depth to the sediment-water interface is often used to estimate the reservoir age. The $^{14}$C age of 13 samples exhibits a good linear relationship with sediments depth in Gahai lake. Extrapolation of this 13 $^{14}$C ages down to the depth of 6 cm yielded a $^{14}$C age of 461 yr BP, while the reliable $^{210}$Pb age at 6 cm is -27 yr BP. Consequently, the difference between the two ages, which amounts to 488 yr, was taken as the reservoir age. Additionally, it's worth noting that independent estimations of the $^{14}$C calibration age and $^{210}$Pb age around 10 cm in Gahai lake was obtained, resulting in values of 497 yr BP and 18 yr BP, respectively. The difference of 479 yr between these two ages can also be considered as the reservoir age. These two methods of estimating reservoir age of Gahai lake show very close, which are mutually supportive. So, the average of 483 yr was adopted as the reservoir age. All original $^{14}$C dates were corrected by subtracting the reservoir age (483 yr) and calibrating them to calendar ages using Calib 8.1. The age-depth model (Fig. 2) was constructed using the Bacon program with the $^{14}$C ages and $^{210}$Pb ages (Blaauw and Andres Christen, 2011) and was reported by Wang et al. (2022)."

**Reference:**

Appleby, P.G., 2002. Chronostratigraphic techniques in recent sediments. In: Tracking Environmental Change Using Lake Sediments. Springer, pp. 171–203.

Blaauw, M., Andres Christen, J., 2011. Flexible Paleoclimate Age-Depth Models Using an Autoregressive Gamma Process. Bayesian Analysis 6, 457-474.

Hou, J.Z., D'Andrea, W.J., Liu, Z.H., 2012. The influence of 14C reservoir age on interpretation of paleolimnological records from the Tibetan Plateau. Quaternary Science Reviews 48, 67-79.

Wang, N., Liu, L., Hou, X., Zhang, Y., Wei, H., Cao, X., 2022. Palynological evidence reveals an arid early Holocene for the northeast Tibetan Plateau. Climate of the Past 18, 2381-2399.

Comment: L39, on "the" TP;
Response: Thanks for the suggestion, we have added "the".

Comment: L47, add "a more" before "rapid warming";
Response: Thanks for the suggestion, we have added "a more".

Comment: L60, I think there are "many" rather than "several" records being published, as you listed in the following sentences.
Response: Thanks for your reminder, we have changed "several" into "many".

Comment: L63, "ice cores" rather than "ice deposits";
Response: Thanks for the suggestion, we have corrected it.

Comment: L92-94, this sentence is a bit confuse, consider to rewrite;
Response: Thanks for the suggestion, we have rephrased this sentence as follows: "Extensive research has been conducted in lakes, employing a single proxy to reconstruct past temperature fluctuations. However, there have been scarce studies that employ various proxies within the same core to reconstruct paleotemperature variations. Furthermore, the limited number of studies primarily concentrate on reconstructing summer temperature and annual average temperature". Please see line 94-99.

Comment: L114, I think the reference "Zhao et al., 2013" is based on alkenones rather than brGDGTs, right?
Response: Thank you very much for your reminder, we have deleted this reference here.

Comment: L117-120, write this sentence, it a bit confuse now;
Response: Thanks for the suggestion, we have rephrased this sentence as follows: "Lake sediments, characterized by their organic matter-rich composition, exhibit continuous and rapid accumulation rates. As a result, they offer high-resolution records of environmental changes, making them highly valued as a primary terrestrial climate archive". Please see line 122-125.

Comment: L143, delete "and it", and replace "which" with "and"; also note Tao "River" and Yellow "River";
Response: Thanks for your suggestion, we have corrected it.

Comment: L148, what is the time/year coverage of the meteorological data? From 1981 to the present, or something else?
Response: Thanks for your meaningful comments. The meteorological data coverage at Langmu Temple station spans from 1957 to 1988. We have added this in the manuscript.

Comment: L149, be specific on how many soil samples, rather than using "several";
Response: Thanks for your reminder. Four catchment soil samples were collected from around the lake. As per your suggestion, we have explicitly mentioned this number (four catchment soils) in the manuscript.

Comment: L175, the more information for chronology should be briefly summarized here as an independent paper;
Response: Thanks for your meaningful comment. We have modified this section, please see line 187-213.
"The chronology of the upper 20 cm of the sediment core is based on measurements of $^{210}$Pb and $^{137}$Cs, at a 1-cm interval. The chronology for the deeper part of the core is provided by accelerator mass spectrometry (AMS) $^{14}$C measurements of 13 bulk sediment samples, which were conducted by Beta Analytic Inc. (Miami, USA) (Fig. 2) (Wang et al., 2022).

The $^{210}$Pb age model was constructed using the constant rate of supply (CRS) model and the $^{137}$Cs peak was used as supplement (Appleby, 2002). The calculated age of $^{210}$Pb using CRS model aligned well with the $^{137}$Cs peak at 6 cm. Overall, the CRS model was deemed suitable for determining the age of Gahai lake.

Reservoir age, as highlighted by Hou et al. (2012), is a crucial factor affecting the age determination of lake sediment cores on the TP. Therefore, it was necessary to establish the reservoir age of Gahai lake before undertaking paleoclimate reconstruction. The linear extrapolation relationship between the $^{14}$C ages and depth to the sediment-water interface is often used to estimate the reservoir age. The $^{14}$C age of 13 samples exhibits a good linear relationship with sediments depth in Gahai lake. Extrapolation of this 13 $^{14}$C ages down to the depth of 6 cm yielded a $^{14}$C age of 461 yr BP, while the reliable $^{210}$Pb age at 6 cm is -27 yr BP. Consequently, the difference between the two ages, which amounts to 488 yr, was taken as the reservoir age. Additionally, it's worth noting that independent estimations of the $^{14}$C calibration age and $^{210}$Pb age around 10 cm in Gahai lake was obtained, resulting in values of 497 yr BP and 18 yr BP, respectively. The difference of 479 yr between these two ages can also be considered as the reservoir age. These two methods of estimating reservoir age of Gahai lake show very close, which are mutually supportive. So, the average of 483 yr was adopted as the reservoir age. All original $^{14}$C dates were corrected by subtracting the reservoir age (483 yr) and calibrating them to calendar ages using Calib 8.1. The age-depth model (Fig. 2) was constructed using the Bacon program with the $^{14}$C ages and $^{210}$Pb ages (Blaauw and Andres Christen, 2011) and was reported by Wang et al. (2022)."

**Reference:**
Appleby, P.G., 2002. Chronostratigraphic techniques in recent sediments. In: Tracking Environmental Change Using Lake Sediments. Springer, pp. 171–203.
Blaauw, M., Andres Christen, J., 2011. Flexible Paleoclimate Age-Depth Models

Using an Autoregressive Gamma Process. Bayesian Analysis 6, 457-474.

Hou, J.Z., D'Andrea, W.J., Liu, Z.H., 2012. The influence of $^{14}$C reservoir age on interpretation of paleolimnological records from the Tibetan Plateau. Quaternary Science Reviews 48, 67-79.

Wang, N., Liu, L., Hou, X., Zhang, Y., Wei, H., Cao, X., 2022. Palynological evidence reveals an arid early Holocene for the northeast Tibetan Plateau. Climate of the Past 18, 2381-2399.

Comment: L215-216, significantly "higher"? double check;

Response: Thank you very much for your reminder, we have corrected it here, it should be "lower".

Comment: L224, replace "abundant" with "abundance";

Response: Thanks for the suggestion, we have corrected this.

Comment: L276, replace "like" with "close to";

Response: Thanks for the suggestion, we have replaced "like" with "close to".

Comment: L285, replace "a" before "new Bayesian" with "the";

Response: Thanks for the reminder, we have replaced "a" with "the" here.

Comment: L298-306, some references here (if there are) would be more helpful. Another possible reason is that the frozen lake surface in winter would insulate the lake water to the atmosphere. Even if there are brGDGTs produced within lake water, they were no longer able to track atmospheric temperature changes during the frozen season (as discussed Sun et al., 2021 and Zhang et al., 2022b as you cited);

Response: Thank you for your suggestion. We have updated the references and modified this section with your meaning suggestion. Below is what we updated, which can also be seen on lines 292-295 and lines 318-323.

Lines 292-295: "Gahai is a shallow lake that is usually completely frozen during winter and spring, and the local meteorological data show that the average snowfall period is 269 days, and that the snowfall period lasts for ~50 days (Luqu County Local Chronicles Compilation Committee, 2006)."

Lines 318-323: "Additionally, the presence of the frozen lake surface during winter creates a thermal barrier, impeding the exchange of heat between the lake water and the atmosphere. Consequently, any brGDGTs generated within the lake water during this period lose their ability to accurately reflect atmospheric temperature variations (Sun et al., 2021; Zhang et al., 2022a). Thus, they were no longer able to track atmospheric temperature changes during the frozen season. So, we prefer to use Gahai brGDGTs to reconstruct temperatures during the summer and ice-free seasons."

**Reference:**

Luqu County Local Chronicles Compilation Committee., 2006. Luqu County Chronicles. Gansu Cultural Publishing House, Lanzhou. pp. 71.

Sun, X., Zhao, C., Zhang, C., Feng, X., Yan, T., Yang, X., Shen, J., 2021. Seasonality in Holocene Temperature Reconstructions in Southwestern China. Paleoceanography and Paleoclimatology 36.

Zhang, C., Zhao, C., Yu, S.-Y., Yang, X., Cheng, J., Zhang, X., Xue, B., Shen, J., Chen, F., 2022a. Seasonal imprint of Holocene temperature reconstruction on the Tibetan Plateau. Earth-Science Reviews 226, 103927.

Comment: L312-314, note the time intervals for BA and YD are different with our current knowledge, this should be briefly discussed;

Response: Your suggestion is very helpful to us. In summary, our records indicate a slight temperature increase during 14.8-11.8 ka, followed by a period of temperature decrease from 11.8-10.5 ka. We propose that these temperature fluctuations may correspond, within the range of dating uncertainties, to the Bølling-Allerød (B/A, 14.8–12.8 ka) and Younger Dryas (YD, 12.8–11.7 ka) events, respectively. Due to the potential presence of age uncertainties, we did not provide detailed elaboration on this aspect in the original text. Additionally, as observed in the Fig. 5a, there is a scarcity of test samples during the 11.8-10.5 ka period. This is attributed to GDGT concentrations falling below the detection limit in these samples. Consequently, we directly connected the reconstructed temperatures at the two points, 11.8 ka and 10.5 ka, resulting in the lowest temperature occurring around 10.5 ka. This deviation in timing introduces a discrepancy with the occurrence of the YD event. We also speculate that climate changes prior to 11.8 ka might have influenced the samples, leading to exceptionally low GDGT concentrations, while the YD event was occurring circa 12.9 ka to 11.7 ka BP. Furthermore, the description provided in our original text may not be accurate, and it is necessary to tone down the assertion of a direct relationship between these two temperature fluctuations and the B/A and YD events. Therefore, we have made the following modifications to this section.

"Within the range of age uncertainties, weak warming occurred during 14.8–11.8 ka, likely corresponding to the Bølling–Allerød (B/A) interstadial. A minor cold reversal occurred during 11.8–10.5 ka, potentially corresponding to the Younger Dryas (YD) event. Notably, the samples collected between 11.8 ka and 10.5 ka exhibited GDGT concentrations below the detection limit. Therefore, we directly linked the temperature reconstructions at the two aforementioned time points, ~11.8 ka and ~10.5 ka, resulting in the lowest temperature of this time period appearing around 10.5 ka. This may cause a time lag with the occurrence of the YD event." Please see lines 378-385.

Comment: L330-333, should mention here you will discuss this in the later section.

Response: Thanks for your suggestion. We have appended a sentence after this

statement, indicating that we will conduct a detailed discussion in the following section. Please see line 415.

Comment: L396-398, why jump to Indian monsoon here? Anything related with your discussion on temperature changes? In particular, the monsoon changes shown by oxygen isotopes from Dongge Cave (as you cited in Figure 6h) do not show a weakened monsoon during the early Holocene. Suggest to delete this sentence.
Response: Thanks for your meaningful comments. We have deleted this sentence.

Comment: L453-456, this statement is not true. In Zhao et al., 2021b (as you cited), they have compared both data using the same calibration, and found a quite similar result.
Response: Your suggestion is very important to us and thanks for your reminder. We have deleted this sentence.

---

## Author Comment (AC2)

Dear Editor and Reviewers:

On behalf of the co-authors, we are very grateful to you for giving us an opportunity to revise our manuscript. We really appreciate your positive and constructive comments together with suggestions on our manuscript entitled 'BrGDGTs-based seasonal paleotemperature reconstruction for the last 15,000 years from a shallow lake on the eastern Tibetan Plateau' (MS No.: cp-2023-32). We have therefore studied reviewers' comments carefully and tried our best to revise our manuscript accordingly. Notably, the changes are highlighted in red in the revised manuscript. Please see below for a point-by-point response to the reviewers' comments and concerns.

**Responds to the comment of Reviewer#2:**

Hou and coworkers have generated a temperature record for the Tibetan Plateau (TP) covering the last 15.000 years. The record is based on branched GDGTs in Lake Gahai, and is compared with other temperature records from the TP to evaluate spatial patterns in the temperature evolution. They find that there are many distinct temperature patterns on the TP and suggest that people need to study their proxy well before interpreting the record.

The paper is clear and well written., but I have some recommendations that will hopefully take the paper one step further.

General recommendations:

The authors currently only discuss the trends in their record, but they do not discuss absolute values, despite the fact that they spend an entire section of the discussion on considering different calibrations. I would like to read more on how the brGDGT-based temperatures relate to the other temperature records. And see the absolute changes in the record interpreted. For example, is the 10 C temperature difference in the record realistic? And why (not)? And if true, what are the implications for our understanding of the climate during the last deglaciation at the TP?

Response: Thanks for your meaningful comments. Our study reconstructed the mean temperature of Months Above Freezing in Gahai lake using a new Bayesian calibration (Martínez-Sosa et al., 2021). The results of this reconstruction indicate temperatures higher than the annual average temperature and lower than the average temperature of summer months (June to August). The average annual temperature in the Gahai region is 1.2°C, and the average temperature during the summer months is 9.9°C. The temperature we reconstructed using surface sediments is 8.8°C, which aligns with the mentioned conditions.

As for the absolute temperature changes since 15,000 yr, although some influential studies indicate a warming of approximately 6.1-7°C from the deglaciation onset to preindustrial times (Tierney et al., 2020; Osman et al., 2021). However, these results are based on global mean sea surface temperatures. Our reconstructed temperature

range is about 10°C, considering the remarkable 'elevation-dependent warming' observed in high-altitude regions compared to low-altitude areas (Mountain Research Initiative EDW Working Group, 2015). Thus, this range could be accurate. Nevertheless, we do not rule out the possibility that our temperature reconstruction may exhibit an overestimation. This is a known issue in temperature reconstruction using biomarkers. Aside from potential uncertainties associated with the biomarkers themselves, calibrations may also significantly influence the observed amplitude. We examined temperature variations reconstructed using different calibrations (Fig. S3), with the smallest range being 6°C and the largest being 12°C. Undoubtedly, further efforts are needed to constrain the inherent uncertainties related to biomarker-based temperature reconstructions.

Our preliminary idea is that using regional or global transfer equations for temperature reconstruction may potentially lead to similar issues. Instead, conducting site-specific calibration for temperature reconstruction may help reduce the amplitude of temperature variability.

**Reference:**

Martínez-Sosa, P., Tierney, J.E., Stefanescu, I.C., Dearing Crampton-Flood, E., Shuman, B.N., Routson, C., 2021. A global Bayesian temperature calibration for lacustrine brGDGTs. Geochimica et Cosmochimica Acta 305, 87-105.

Mountain Initiative EDW Working Group, 2015. Elevation-dependent warming in mountain regions of the world. Nature Climate Change 5, 424-430.

Osman, M.B., Tierney, J.E., Zhu, J., Tardif, R., Hakim, G.J., King, J., Poulsen, C.J., 2021. Globally resolved surface temperatures since the Last Glacial Maximum. Nature 599, 239-244.

Tierney, J.E., Zhu, J., King, J., Malevich, S.B., Hakim, G.J., Poulsen, C.J., 2020. Glacial cooling and climate sensitivity revisited. Nature 584, 569-+.

Comment: As mentioned in the previous comment there is a lot of discussion on which calibration to use, but in the end the currently available and widely-accepted calibrations (i.e., the ones that use both 5-methyl and 6-methyl brGDGTs) are all based on MBT'5me and will thus generate the same record, except for the values on the temperature axis. Instead of discussing all the possible calibrations, including those based on outdated chromatography methods, or on lake systems that are not comparable to the one targeted here (e.g., saline lakes, East African Lakes with limited seasonality), I would rather like to see more discussion on the drivers of brGDGT production in this specific lake. In my opinion it is clear that brGDGTs in lakes are mostly produced in situ, and are sensitive to O2 availability and redox conditions, which in turn is often related to the mixing regime, trophic state, and/or depth of the lake (e.g., Weber et al., 2018 PNAS, van Bree et al., 2020 Biogeosciences, Wu et al., 2021 Chem. Geol). A good understanding of where and

when brGDGTs are produced in the lake, and how this is related to temperature, provides support for the selection of the 'right' transfer function.

Response: We thank Reviewer for the meaningful comments and suggestions. We have made revisions to this section based on the reviewers'2# comments, mainly emphasizing the specific status of lake Gahai and elaborating on our utilization of the calibration from Martínez-Sosa et al. (2021).

For a detailed description, please refer to the following introduction.

[revised manuscript text omitted]

Comment: Explain the drivers of the current and past climate on the TP. What are the most important wind systems? I tend to see it as influenced by the East Asian Monsoon system, but this is nowhere mentioned. In contrary, there is an out-of-the-blue reference to the Indian Summer Monsoon in L397, but an explanation of its connection to the climate on the TP is not provided.

Response: Thanks for your meaningful comments. We have deleted this sentence "It is possible that these factors impacted the summer temperatures in the Indian Summer Monsoon (ISM) domain via ocean-atmosphere interactions". And we have rephased the part in lines 422-426 as follows "The brGDGTs-based temperature record from Gahai confirms the occurrence of a climate optimum in the mid-Holocene on the northeast Tibetan Plateau, which is consistent with several other pollen and pollen-reconstructed temperature records from the fringe areas of the Asian summer

monsoon (Fig. 6), suggesting that it is a reliable representation of Holocene temperature changes in this region."

Other comments:

Comment: Introduction: The rationale on which exact temperature different proxies represent is not well explained. For example, the chironomid record from Tiancai lake is presented as July temperature. How valid is this? The growing season of chironomids may change over time, especially over glacial/interglacial transitions. Is it always July? The same is true for brGDGTs and pollen. Do they present annual or seasonal (and in that case: ice-free, or summer, or growing season) temperatures? How can we tell, and can we assume that seasonality is constant over time? This is important because climate modelers will want to use these records and it is our responsibility to make sure that they use the right temperatures to warrant low uncertainty on model projections of future climate. For example, the set of aims listed at the end of the introduction raises the question if ice-free season temperatures equal a warm-bias? And why compare the brGDGT record with July temperatures? What is the reason behind that?

Response: Thank you for your suggestion. The issue you mentioned is of paramount importance and remains an ongoing challenge yet to be resolved. The method of transferring contemporary knowledge and principles to the study of the past, known as the 'present is the key to the past,' stands as a foundational and paramount methodology within the realm of Earth sciences. This approach involves deducing the conditions, processes, and characteristics of geological events in ancient times based on contemporary understanding and principles.

While the application of current knowledge to the study of the past is viable to a certain extent, it bears the potential for inadvertent deviations from factual precision. Consequently, addressing this issue in the context of the study of lacustrine sediments from the Holocene epoch requires an initial assessment of sedimentary environment stability based on factors such as existing sedimentation rates, chronology, lithological variations, and more. This evaluation helps prevent significant sedimentary discontinuities or other impactful events that could affect the reliability of proxy indicators.

The mentioned study on chironomid record from Tiancai lake, as referenced in Zhang et al., 2017a, explicitly demonstrates that chironomid respond to July temperatures. The chironomid-based transfer function, developed from the region of southwestern China based on a 100-lake calibration training set (Zhang et al., 2017b) was employed to translate the chironomid assemblage data from Tiancai lake into a reconstruction of mean July temperatures. This transfer function, constructed via a weighted averaging partial least squares (WA-PLS) regression in C2 (Juggins, 2005), spans a range of mean July temperature from 4.2 to 20.8 °C, whereas Tiancai lake currently exhibits a modern

mean July temperature value of 8.3 ℃. This modern process has been applied to the Tiancai lake sediment core, thus enabling the reconstruction of July temperatures over the past 9,000 years. The original authors consider this reconstruction to be reliable.

Similarly, through the examination of modern processes involving global lake and soil, it has been concluded that both brGDGTs and pollen respond to temperature changes. Consequently, these indicators have been employed extensively for temperature reconstructions in Quaternary paleoclimate research, embracing the 'present is the key to the past' principle. However, questions regarding the consistent existence of seasonality remain unresolved. This underscores one of the limitations of paleoclimate reconstruction work, motivating a clear understanding of the indicative significance, sensitivity, and representativeness of relevant proxy indicators. This will help mitigate uncertainties and provide more accurate records for paleoclimate simulation studies.

Finally, temperatures during the non-freezing seasons tend to exceed annual averages yet remain lower than summer temperatures. We are inclined to attribute this to warm-season bias, as opposed to a warm bias. As for why brGDGT-based temperature reconstructions are compared to July temperatures, one prevailing hypothesis addressing the "Holocene temperature conundrum" highlights the possible role of seasonal discrepancies in proxy indicators. These discrepancies could arise from varying climatic implications intrinsic to the proxy indicators themselves. Therefore, a comprehensive understanding of the climatic significance, sensitivity, and representativeness of proxy indicators is pivotal. Undertaking multi-proxy temperature reconstructions within the same study area could shed light on potential seasonal deviations in different temperature series and contribute to unraveling the "Holocene temperature conundrum". Presently, the bulk of studies rely on single temperature proxies. In contrast, we have utilized two distinct temperature proxies, brGDGTs and pollen, from the same borehole to reconstruct Holocene temperature, thus vividly illustrating the existence of seasonal biases in different indicators.


Comment: Calibration chosen. See also my main comment. I suggest to change the argumentation. Rather than applying all calibrations ever published and picking the one that fits best, present a rationale for using the one calibration that is most suitable for this lake /dataset and then apply only that one.

Response: Thanks for your meaningful comments. It's very helpful to us. We have made detailed revisions to this section based on the reviewer's comments, as evidenced in lines 292-372.

[revised manuscript text omitted]

Comment: The authors present their record as "warm-biased temperature record". I find this not so informative, as it is not clear why it has a bias and where it comes from. I suggest to specify the bias further so outsides know what the record represents and how they can use it in future studies.

Response: Thank you for your suggestion. In this study, we have reconstructed the mean temperature of Months Above Freezing. We have already replaced 'warm-biased' with 'the mean temperature of Months Above Freezing' in certain sections.

Comment: Questions I had while reading the paper: how is it so clear that the record contains a warm bias? In the end, the calibration from Martinez-Sosa et al 2021 is used, but this transfer function uses mean air temperature for months above freezing (MAF). In other words, the record that is presented is a record of MAF, and thus by default warmer than MAT. If there are any further offsets or structural biases from MAF then they have to be identified and explained in the text.

Response: Thank you for your suggestions. As you mentioned, we have employed the calibration proposed by Martinez-Sosa et al. (2021) in this study to reconstruct temperatures during the ice-free seasons. Without a doubt, temperatures during ice-free seasons are higher than the annual average temperatures. Therefore, we believe that the reconstructed records in this paper contain a certain warm bias.

As detailed in the section about study site, Gahai lake is located on the eastern edge of the Tibetan Plateau, with consistently low temperatures throughout the year. The maximum depth of the lake does not exceed two meters. From October to April in the next year, the lake is predominantly covered by ice. Consequently, the sediment cores likely record the average temperatures during the ice-free months. Furthermore, Gahai lake is presently an open lake, with its waters flowing northwestward through the Tao River and eventually into the Yellow River. Thus, we have reason to believe that the lake's water level has not exceeded two meters since the Holocene. Given the shallowness of the lake in such a cold region, we posit that brGDGTs in the lake sediment are more likely to grow during the warmer seasons, while the complete freezing of the lake during ice-cover periods restricts the growth of brGDGT producers.

Comment: How can brGDGTs represent both annual mean air temperature and a warm-biased temperature (L35-36 in the abstract)?

Response: Thank you for your reminder. Here, we are referring to instances where brGDGTs are utilized for various interpretations in different studies. For example, in some articles, brGDGTs are considered to reflect annual average temperatures, while in others, they are believed to indicate warm-biased temperatures. This might have caused ambiguity in our original statement. Thus, we have rephrased the sentence for clarity.

"The results demonstrate that brGDGTs have been employed to reconstruct various temperatures in different studies, including annual average temperature and warm-biased temperature." Please see lines 35-37.

Comment: L169: add number of soils sampled and indicate the depth interval, and also plot them on the triplot (Fig. 4). Actually, it is quite clear that such a triplot does not really separate brGDGTs in lake sediments from soils on a global scale – it seems best suited to identify in situ production in the marine coastal zone (Sinninghe Damsté, 2016). However, lakes tend to have more hexamethylated brGDGTs (in

particular IIIa), so directly comparing the relative abundances of brGDGTs and soils with a bar-plot could be more informative (see e.g., Fig. 11 in Martinez-Sosa et al., 2021). It can also help to do Principle Component Analysis (PCA) on the lake sediment record, and then passively add the soils to see where they plot. The sample distribution in the PCA can further help to identify the main changes in brGDGT distributions through time and possibly connect this to likely environmental drivers (and finally temperature).

Response: Thanks for your suggestions. Four catchment soil samples were collected from around the lake. As per your suggestion, we have explicitly mentioned this number (four catchment soils) in the manuscript and plot them on the triplot (see Fig. 1). In addition, we also compared the relative abundance of brGDGTs in soils and core sediment samples with a bar-plot in Fig 2 here, which is also presented in Supplementary Fig. S2.

[Figure]

Fig. 1 Comparison of the fractional abundances of tetramethylated, pentamethylated, and hexamethylated bGDGTs in sediment core samples and catchment soils from Gahai with lake surface sediments from the Tibetan Plateau (Günther et al., 2014; Wang et al., 2016), East Africa (Russell et al., 2018), and worldwide (Martínez-Sosa et al., 2021).

[Figure]

Fig. 2 Mean fractional abundances and standard deviations of brGDGTs from downcore sediments and catchment soil samples in Gahai lake.

Comment: The methods can do with a bit more information on the lithology and the age model. In the Discussion the oldest 5000 years are suddenly discarded (L308), as these sediments could represent a slump, but no evidence is provided.

Response: We are very grateful to you for your meaningful comments. We have included this part in the text, please see line 187-213.

"The chronology of the upper 20 cm of the sediment core is based on measurements of $^{210}$Pb and $^{137}$Cs, at a 1-cm interval. The chronology for the deeper part of the core is provided by accelerator mass spectrometry (AMS) $^{14}$C measurements of 13 bulk sediment samples, which were conducted by Beta Analytic Inc. (Miami, USA) (Fig. 2) (Wang et al., 2022).

The $^{210}$Pb age model was constructed using the constant rate of supply (CRS) model and the $^{137}$Cs peak was used as supplement (Appleby, 2002). The calculated age of $^{210}$Pb using CRS model aligned well with the $^{137}$Cs peak at 6 cm. Overall, the CRS model was deemed suitable for determining the age of Gahai lake.

Reservoir age, as highlighted by Hou et al. (2012), is a crucial factor affecting the age determination of lake sediment cores on the TP. Therefore, it was necessary to establish the reservoir age of Gahai lake before undertaking paleoclimate reconstruction. The linear extrapolation relationship between the $^{14}$C ages and depth to the sediment-water interface is often used to estimate the reservoir age. The $^{14}$C age of 13 samples exhibits a good linear relationship with sediments depth in Gahai lake. Extrapolation of this 13 $^{14}$C ages down to the depth of 6 cm yielded a $^{14}$C age of 461 yr BP, while the reliable $^{210}$Pb age at 6 cm is -27 yr BP. Consequently, the difference between the two ages, which amounts to 488 yr, was taken as the reservoir age. Additionally, it's worth noting that independent estimations of the $^{14}$C calibration age and $^{210}$Pb age around 10 cm in Gahai lake was obtained, resulting in values of 497 yr BP and 18 yr BP, respectively.

The difference of 479 yr between these two ages can also be considered as the reservoir age. These two methods of estimating reservoir age of Gahai lake show very close, which are mutually supportive. So, the average of 483 yr was adopted as the reservoir age. All original [14]C dates were corrected by subtracting the reservoir age (483 yr) and calibrating them to calendar ages using Calib 8.1. The age-depth model (Fig. 2) was constructed using the Bacon program with the [14]C ages and [210]Pb ages (Blaauw and Andres Christen, 2011) and was reported by Wang et al. (2022)."

Regarding the lithology, especially the sudden exclusion of the oldest 5000 years, we believe that the corresponding time interval (from 191 cm) may have undergone rapid sedimentation or alternatively slumping. There are several reasons for this:
1. We can observe that the ages around 191 cm, 229 cm and 279 cm are relatively close (15070 a, 14870 a, 15500 a, respectively), which suggests the possibility of rapid sedimentation.
2. By examining the grain size distribution, we can notice significant fluctuations in the silt fraction (4-63 μm) starting from 191 cm (Fig. 3). The silt fraction in Gahai is driven by the medium silt (16-32 μm) fraction, while the fine and coarse silt fractions remain almost unchanged during the Holocene, hence the fine, medium, and coarse silts are combined into the total silt fraction (4-63 μm) for discussion (Wang et al., 2022). This could indicate the occurrence of specific events.

[Figure]

Fig. 3 Silt fraction (4-63 μm) distribution.

3. During this stage, the concentration of brGDGTsis notably low, which hinders our ability to conduct thorough analysis. Similarly, the pollen concentration during this time period is also quite low, and the data are insufficient for statistical analyses (Wang et al., 2022).

Therefore, we only present the research results from the past 15,000 years.

Comment: Interpretation of the temperature record by vegetation change in the area, and thus heat capacity (L333): Note that this process is mostly valid for soils, for which this has been described, but that the vast majority (if not all) of the brGDGTs in the lake record will be produced in situ. Assuming that the lake has always contained water during the studied interval, the heat capacity would not have changed.

Response: Thank you very much for your suggestion, and we have removed this statement.

Remediation effects: Since this anthropogenic spike only represents a few datapoints I would be careful not to overinterpret the data. Especially stating that remediation effects have an effect and that brGDGTs now record temperature again based on only the surface sediment would be overstating it for me. Please tone down and substantially reduce the discussion on this part of the record.

Response: Thank you for your reminder. In accordance with your suggestion, we have reduced the discussion in this section.

Minor comments:

Comment: Check the order of references. By first name, then by year.

Response: Thank you for your suggestions, we have made individual revisions throughout the manuscript.

Comment: Often use of 'significant' without providing p-value.

Response: Thank you for your reminder, we have thoroughly checked and made corresponding revisions to each of them.

Comment: Damsté et al should be Sinninghe Damsté et al.

Response: Thank you very much for your reminder. We have conducted a thorough review and made all the necessary modifications.

Comment: Provide information on the detection limit used for GDGT quantification.

Response: Thank you for your suggestion. During the sample testing process, the detection limit for GDGTs testing is 0.0004 ppm.

---

## Author Response (AR3)

From: Dr. Juzhi Hou
Institute of Tibetan Plateau Research
Chinese Academy of Sciences

Dec 19, 2023

To: Dr. Qiuzhen Yin
Editor
Climate of the Past

Dear Dr. Yin,

Thank you very much for your comments and suggestion on our manuscript entitled "BrGDGTs-based seasonal paleotemperature reconstruction for the last 15,000 years from a shallow lake on the eastern Tibetan Plateau" (No. cp-2023-32). We have carefully addressed your comments point-to-point below and highlighted in the text. We wish the revised manuscript meets the publication criteria for Climate of the Past.

**Comments:**
Line 36-38: This sentence is not clear. "a warming trend during the Holocene" is told at the beginning, but "a cooling trend" is said at the end.

**Response:** Thank you for your suggestion. We have rephrased the sentence, "The existing/available temperature records show complicated patterns of variation, some with general warming trends throughout the Holocene, some with cooling trends, while some with warm middle Holocene".

The abstract is too long and sometimes confusing. I suggest to make it more concise.

**Response:** Thank you for your suggestion. We have shorten the abstract, please see the revised text.

Line 430-431: Apparently this is not what Fig.6 tells. this sentence needs to be modified.

**Response:** We appreciate the comment and we have deleted this sentence, which isn't suitable here. Additionally, we have merged the next paragraph with the previous one.

Figure 7: What are the red and blue curves? How is this Jun-Sep insolation calculated and what is the unit? I tried different ways to make the calculation, but I could not get similar result as the authors. What is the Jul insolation and its unit? monthly mean, mid-month daily insolation, total insolation? Why do you need to show both insolation curves?

**Response:** Thank you very much for your advice and assistance. We have clarified the confusion in Fig. 7. The red curve represents the cumulative insolation from June to September in $W/m^2$, and the navy blue curve shows the mean insolation during July in $W/m^2$. Both are shown for comparison with temperature reconstruction in ice-free season (brGDGTs in this study) and in July (pollen in Wang et al., (2022)) at Lake Gahai. The difference in the two temperature records may result from the lags between the insolation in June to September and in July.

The previous solar radiation data was obtained from https://vo.imcce.fr/insola/earth/online/earth/online/index.php. The 34°N July mean insolation and June-September mean insolation calculated in Fig.7 are obtained by dividing the total solar radiation for July 1st to July 31st and June 1st to September 30th, respectively, by the length of these two time periods. The total solar radiation is calculated using the Berger et al. (2010) elliptic integral method.

We have already labeled this in the caption of Fig. 7f, named "Mean insolation during July (W/m2) (navy blue curve) and mean insolation during ice-free months (W/m2) at 34 °N (red curve)". Relevant references have been cited.

References:

Berger, A. and Loutre, M. F.: Insolation values for the climate of the last 10000000 years, Quaternary Science Reviews, 10, 297-317, 10.1016/0277-3791(91)90033-q, 1991.

Berger, A., Loutre, M. F., and Yin, Q.: Total irradiation during any time interval of the year using elliptic integrals, Quaternary Science Reviews, 29, 1968-1982, 10.1016/j.quascirev.2010.05.007, 2010.

Wang, N., Liu, L., Hou, X., Zhang, Y., Wei, H., and Cao, X.: Palynological evidence reveals an arid early Holocene for the northeast Tibetan Plateau, Climate of the Past, 18, 2381-2399, 10.5194/cp-18-2381-2022, 2022.

L447-453: The ice sheets during early Holocene would cool the system directly through their cooling effect, not necessarily through melting water effect. If you suggest melting water effect, you better compare your records with melt water flux. How about the effect of $CO_2$?
About the effects of astronomical parameters, ice sheets and $CO_2$ on East Asia climate, I recommend this paper https://doi.org/10.1016/j.quascirev.2022.107689

**Response:** We thank editor for the meaningful comments and suggestions. We briefly described the impact of $CO_2$ on temperature changes and rephrased this part in the revision; see lines 450-452 and 458-462 for details.

Line 450-452: "Furthermore, the cooling during the early Holocene followed by the warming trend in the mid-Holocene potentially correlates with significant fluctuations in $CO_2$ concentrations within these intervals (Fig. 7e) (Monnin et al., 2004)."

Line 458-462: "In essence, temperature, especially seasonal variations like the Gahai ice-freeze temperature in the eastern TP, is influenced by multifaceted factors including astronomical forcing, $CO_2$, and ice sheets. Temperature exhibits varied sensitivities in response to these factors, while both insolation and $CO_2$ exert considerable and favorable impacts on summer temperature patterns (Lyu and Yin, 2022)."

References:

Lyu, A. and Yin, Q. Z.: The spatial-temporal patterns of East Asian climate in response to insolation, $CO_2$ and ice sheets during MIS-5, Quaternary Science Reviews, 293, 10.1016/j.quascirev.2022.107689, 2022.

Monnin, E., Steig, E. J., Siegenthaler, U., Kawamura, K., Schwander, J., Stauffer, B., Stocker, T. F., Morse, D. L., Barnola, J. M., Bellier, B., Raynaud, D., and Fischer, H.: Evidence for substantial accumulation rate variability in Antarctica during the Holocene, through synchronization of CO2 in the Taylor Dome, Dome C and DML ice cores, Earth and Planetary Science Letters, 224, 45-54, 10.1016/j.epsl.2004.05.007, 2004.

I am still wondering why pollen record reflects only July temperature, why not the warm season temperature.

**Response:** Thank you for your insightful comments. The referenced pollen record from Lake Gahai, as outlined in Wang et al., 2022, explicitly demonstrates the responsiveness of pollen assemblages to the mean temperature of the warmest month (July). A pollen-based transfer function was developed for the eastern TP, utilizing 117 lake surface sediment samples (n = 117) obtained between elevations of 3720 and 5170 meters above sea level (Cao et al., 2021). This function was calibrated with modern climatic data sourced from the Chinese Meteorological Forcing Dataset.

In their study, Wang et al. (2022) specifically selected the mean temperature of July along with elevation (Elev) to explore the relationship between pollen assemblages and July temperature. Ordination analysis highlighted the significance of July temperature as a climatic determinant affecting pollen distribution. Calibration sets, linking pollen data with July temperature, were established to evaluate the predictive power of this dataset. The 'leave-one-out' cross-validation results indicated strong performance, particularly with the first component for July temperature. The original authors therefore regard this reconstruction as reliable.

To assess the potential of the pollen dataset for reconstructing past climates, Weighted Averaging Partial Least Squares Regression (WA-PLS) was employed. Its performance was tested using 'leave-one-out' cross-validation (ter Braak & Juggins, 1993) and evaluated through $R^2$ (coefficient of determination between observed and predicted values) and RMSEP (root mean square error of prediction Birks, 1998). The

number of WA-PLS components used was determined via a randomization t-test (Juggins and Birks, 2012). Climate reconstruction was conducted using R software with the rioja package version 0.7-3 (Juggins, 2012). Prior to reconstruction, the pollen assemblages underwent square-root transformation to minimize noise (Prentice, 1980).

After discussing with the original authors, they believe that the temperature reconstructed for the Gahai region corresponds to the temperature in July.

References:

Cao, X., Tian, F., Li, K., Ni, J., Yu, X., Liu, L. and Wang, N. N.: Lak e surf ace sediment pollen dataset for the alpine meadow vegetation type from the eastern Tibetan Plateau and its potential in past climate reconstructions Earth Syst Sci Data 13, 3525-3537, https://doi.org/ 10.5194/essd 13-3525-2021, 2021.

Juggins, S.: Rioja: analysis of Quaternary Science Data ve rs ion 0.7-3, available at: http://cran.r-project.org/web/packages/rioja/ index.html (last access: June 2020), 2012.

Juggins, S. and Birks, H. J. B.: Quantitative environmental reconstructions from biological data, in: Birks, H. J. B., Lotter, A. F., Juggins, S., and Smol, J. P., Tracking environmental change using lake sediments (vol 5): Data handling and 91 numerical techniques, Springer, Dordrecht, 431-494, 2012.

Prentice, I. C.: Multidimensional scaling as a research tool in Quaternary palynology: a review of theory and methods, Rev. Palaeobot. Palyno., 31, 71–104, https://doi.org/10.1016/0034-6667(80)90023-8, 1980.

ter Braak, J. F., and Juggins, S: Weighted averaging partial least squares regression (WA-PLS): an improved method for reconstructing environmental variables from species assemblages Hydrobiologia 269-270, 485-502 doi: 10.1007/BF00028046, 1993.

Wang, N., Liu, L., Hou, X., Zhang, Y., Wei, H., and Cao, X.: Palynological evidence reveals an arid early Holocene for the northeast Tibetan Plateau, Climate of the Past, 18, 2381-2399, 10.5194/cp-18-2381-2022, 2022.

I draw the authors' attention that Copernicus Publications request depositing data that correspond to journal articles in reliable (public) data repositories. Please see the data policy at https://www.climate-of-the-past.net/policies/data_policy.html.

**Response:** Thanks for your reminder. We have prepared the data and will submit it as requested.

---

## Author Response (AR4)

From: Dr. Juzhi Hou
Institute of Tibetan Plateau Research
Chinese Academy of Sciences

9/18/2023
To: Dr. Qiuzhen Yin
Editor
Climate of the Past

Dear Dr. Yin,
On behalf of the co-authors, we are very grateful to you for giving us an opportunity to revise our manuscript. We really appreciate your positive and constructive comments together with suggestions on our manuscript entitled 'BrGDGTs-based seasonal paleotemperature reconstruction for the last 15,000 years from a shallow lake on the eastern Tibetan Plateau' (MS No.: cp-2023-32). We have therefore studied reviewers' comments carefully and tried our best to revise our manuscript accordingly. Notably, the changes are highlighted in red in the revised manuscript. Please see below for a point-by-point response to the reviewers' comments and concerns.

**Editor and Reviewer comments:**
**Reviewer#1:**
The potential seasonal bias produced by terrestrial archives are important to better understand the so-called "Holocene Temperature Conundrum", the difference between simulated global Holocene warming and proxy-reconstructed global Holocene cooling. In this manuscript, Hou et al analyze down-core brGDGTs from a sediment core collected from Gahai Lake on the eastern Tibetan Plateau. Based on brGDGTs, they reconstruct an ice-free season (May to September) temperature over the past 15 ka. They also compare the new record with a previously published pollen-based July temperature record from the same core, and find that Holocene Thermal Maximum in the ice-free season record lags that in the July temperature record. They also review other published brGDGT-based temperature records, and evaluate differences among them. They emphasize the importance of considering lake conditions and modern-process investigations when using brGDGTs to reconstruct past climate changes.

Overall, I think this study provide some valuable data to better understanding the seasonal bias in Holocene temperature reconstruction. In particular, difference records from the same sediment core from the same lake add the credibility of the results. Moreover, the systematic modern process analyzes for brGDGTs sources in lake catchment basin significantly improve the quality of the reconstructed record. Therefore, I would recommend a minor revision.

One main issue the authors should be considered is the chronology. They author simply cited the age-model results from a published paper, without any detailed explanation. For my understanding, as an independent paper, the author should necessarily and

concisely explain how they reconstructed the age model and how they evaluate the potential "old carbon" effect. Although these potential age uncertainties won't affect the main finding for Holocene climate changes, they seem do affect the timing of the deglacial BA and YD events.

**Response:** We are very grateful to you for your meaningful comments. As you said, as an independent article, it should indeed have a detailed introduction to the chronology. We have included this part in the text, please see lines 187-213.

"The chronology of the upper 20 cm of the sediment core is based on measurements of $^{210}$Pb and $^{137}$Cs, at a 1-cm interval. The chronology for the deeper part of the core is provided by accelerator mass spectrometry (AMS) $^{14}$C measurements of 13 bulk sediment samples, which were conducted by Beta Analytic Inc. (Miami, USA) (Fig. 2) (Wang et al., 2022).

The $^{210}$Pb age model was constructed using the constant rate of supply (CRS) model and the $^{137}$Cs peak was used as supplement (Appleby, 2002). The calculated age of $^{210}$Pb using CRS model aligned well with the $^{137}$Cs peak at 6 cm. Overall, the CRS model was deemed suitable for determining the age of Gahai lake.

Reservoir age, as highlighted by Hou et al. (2012), is a crucial factor affecting the age determination of lake sediment cores on the TP. Therefore, it was necessary to establish the reservoir age of Gahai lake before undertaking paleoclimate reconstruction. The linear extrapolation relationship between the $^{14}$C ages and depth to the sediment-water interface is often used to estimate the reservoir age. The $^{14}$C age of 13 samples exhibits a good linear relationship with sediments depth in Gahai lake. Extrapolation of this 13 $^{14}$C ages down to the depth of 6 cm yielded a $^{14}$C age of 461 yr BP, while the reliable $^{210}$Pb age at 6 cm is -27 yr BP. Consequently, the difference between the two ages, which amounts to 488 yr, was taken as the reservoir age. Additionally, it's worth noting that independent estimations of the $^{14}$C calibration age and $^{210}$Pb age around 10 cm in Gahai lake was obtained, resulting in values of 497 yr BP and 18 yr BP, respectively. The difference of 479 yr between these two ages can also be considered as the reservoir age. These two methods of estimating reservoir age of Gahai lake show very close, which are mutually supportive. So, the average of 483 yr was adopted as the reservoir age. All original $^{14}$C dates were corrected by subtracting the reservoir age (483 yr) and calibrating them to calendar ages using Calib 8.1. The age-depth model (Fig. 2) was constructed using the Bacon program with the $^{14}$C ages and $^{210}$Pb ages (Blaauw and Andres Christen, 2011) and was reported by Wang et al. (2022)."

Reference:

Appleby, P.G., 2002. Chronostratigraphic techniques in recent sediments. In: Tracking Environmental Change Using Lake Sediments. Springer, pp. 171–203.

Blaauw, M., Andres Christen, J., 2011. Flexible Paleoclimate Age-Depth Models Using an Autoregressive Gamma Process. Bayesian Analysis 6, 457-474.

Hou, J.Z., D'Andrea, W.J., Liu, Z.H., 2012. The influence of 14C reservoir age on interpretation of paleolimnological records from the Tibetan Plateau. Quaternary Science Reviews 48, 67-79.

Wang, N., Liu, L., Hou, X., Zhang, Y., Wei, H., Cao, X., 2022. Palynological evidence

reveals an arid early Holocene for the northeast Tibetan Plateau. Climate of the Past 18, 2381-2399.

Comment: L39, on "the" TP;
Response: Thanks for the suggestion, we have added "the".

Comment: L47, add "a more" before "rapid warming";
Response: Thanks for the suggestion, we have added "a more".

Comment: L60, I think there are "many" rather than "several" records being published, as you listed in the following sentences.
Response: Thanks for your reminder, we have changed "several" into "many".

Comment: L63, "ice cores" rather than "ice deposits";
Response: Thanks for the suggestion, we have corrected it.

Comment: L92-94, this sentence is a bit confuse, consider to rewrite;
Response: Thanks for the suggestion, we have rephrased this sentence as follows: "Extensive research has been conducted in lakes, employing a single proxy to reconstruct past temperature fluctuations. However, there have been scarce studies that employ various proxies within the same core to reconstruct paleotemperature variations. Furthermore, the limited number of studies primarily concentrate on reconstructing summer temperature and annual average temperature". Please see line 94-99.

Comment: L114, I think the reference "Zhao et al., 2013" is based on alkenones rather than brGDGTs, right?
Response: Thank you very much for your reminder, we have deleted this reference here.

Comment: L117-120, write this sentence, it a bit confuse now;
Response: Thanks for the suggestion, we have rephrased this sentence as follows: "Lake sediments, characterized by their organic matter-rich composition, exhibit continuous and rapid accumulation rates. As a result, they offer high-resolution records of environmental changes, making them highly valued as a primary terrestrial climate archive". Please see lines 122-125.

Comment: L143, delete "and it", and replace "which" with "and"; also note Tao "River" and Yellow "River";
Response: Thanks for your suggestion, we have corrected it.

Comment: L148, what is the time/year coverage of the meteorological data? From 1981 to the present, or something else?
Response: Thanks for your meaningful comments. The meteorological data coverage at Langmu Temple station spans from 1957 to 1988. We have added this in the

manuscript.

Comment: L149, be specific on how many soil samples, rather than using "several";
**Response:** Thanks for your reminder. Four catchment soil samples were collected from around the lake. As per your suggestion, we have explicitly mentioned this number (four catchment soils) in the manuscript.

Comment: L175, the more information for chronology should be briefly summarized here as an independent paper;
**Response:** Thanks for your meaningful comment. We have modified this section, please see line 187-213.

"The chronology of the upper 20 cm of the sediment core is based on measurements of $^{210}$Pb and $^{137}$Cs, at a 1-cm interval. The chronology for the deeper part of the core is provided by accelerator mass spectrometry (AMS) $^{14}$C measurements of 13 bulk sediment samples, which were conducted by Beta Analytic Inc. (Miami, USA) (Fig. 2) (Wang et al., 2022).

The $^{210}$Pb age model was constructed using the constant rate of supply (CRS) model and the $^{137}$Cs peak was used as supplement (Appleby, 2002). The calculated age of $^{210}$Pb using CRS model aligned well with the $^{137}$Cs peak at 6 cm. Overall, the CRS model was deemed suitable for determining the age of Gahai lake.

Reservoir age, as highlighted by Hou et al. (2012), is a crucial factor affecting the age determination of lake sediment cores on the TP. Therefore, it was necessary to establish the reservoir age of Gahai lake before undertaking paleoclimate reconstruction. The linear extrapolation relationship between the $^{14}$C ages and depth to the sediment-water interface is often used to estimate the reservoir age. The $^{14}$C age of 13 samples exhibits a good linear relationship with sediments depth in Gahai lake. Extrapolation of this 13 $^{14}$C ages down to the depth of 6 cm yielded a $^{14}$C age of 461 yr BP, while the reliable $^{210}$Pb age at 6 cm is -27 yr BP. Consequently, the difference between the two ages, which amounts to 488 yr, was taken as the reservoir age. Additionally, it's worth noting that independent estimations of the $^{14}$C calibration age and $^{210}$Pb age around 10 cm in Gahai lake was obtained, resulting in values of 497 yr BP and 18 yr BP, respectively. The difference of 479 yr between these two ages can also be considered as the reservoir age. These two methods of estimating reservoir age of Gahai lake show very close, which are mutually supportive. So, the average of 483 yr was adopted as the reservoir age. All original $^{14}$C dates were corrected by subtracting the reservoir age (483 yr) and calibrating them to calendar ages using Calib 8.1. The age-depth model (Fig. 2) was constructed using the Bacon program with the $^{14}$C ages and $^{210}$Pb ages (Blaauw and Andres Christen, 2011) and was reported by Wang et al. (2022)."


Comment: L312-314, note the time intervals for BA and YD are different with our current knowledge, this should be briefly discussed;

**Response:** Your suggestion is very helpful to us. In summary, our records indicate a slight temperature increase during 14.8-11.8 ka, followed by a period of temperature decrease from 11.8-10.5 ka. We propose that these temperature fluctuations may correspond, within the range of dating uncertainties, to the Bølling-Allerød (B/A, 14.8–12.8 ka) and Younger Dryas (YD, 12.8–11.7 ka) events, respectively. Due to the potential presence of age uncertainties, we did not provide detailed elaboration on this aspect in the original text. Additionally, as observed in the Fig. 5a, there is a scarcity of test samples during the 11.8-10.5 ka period. This is attributed to GDGT concentrations falling below the detection limit in these samples. Consequently, we directly connected the reconstructed temperatures at the two points, 11.8 ka and 10.5 ka, resulting in the lowest temperature occurring around 10.5 ka. This deviation in timing introduces a discrepancy with the occurrence of the YD event. We also speculate that climate changes prior to 11.8 ka might have influenced the samples, leading to exceptionally low GDGT concentrations, while the YD event was occurring circa 12.9 ka to 11.7 ka BP. Furthermore, the description provided in our original text may not be accurate, and it is necessary to tone down the assertion of a direct relationship between these two temperature fluctuations and the B/A and YD events. Therefore, we have made the following modifications to this section.

"Within the range of age uncertainties, weak warming occurred during 14.8–11.8 ka, likely corresponding to the Bølling–Allerød (B/A) interstadial. A minor cold reversal occurred during 11.8–10.5 ka, potentially corresponding to the Younger Dryas (YD) event. Notably, the samples collected between 11.8 ka and 10.5 ka exhibited GDGT concentrations below the detection limit. Therefore, we directly linked the temperature reconstructions at the two aforementioned time points, ~11.8 ka and ~10.5 ka, resulting in the lowest temperature of this time period appearing around 10.5 ka. This may cause a time lag with the occurrence of the YD event."

Please see lines 378-385.

Comment: L330-333, should mention here you will discuss this in the later section.

**Response:** Thanks for your suggestion. We have appended a sentence after this statement, indicating that we will conduct a detailed discussion in the following section.

Please see line 415.

**Response:** Thanks for your meaningful comments. We have deleted this sentence.

**Response:** We appreciate reviewer to point it out, and we have deleted this sentence in the revision.

**Reviewer#2:**
Hou and coworkers have generated a temperature record for the Tibetan Plateau (TP) covering the last 15.000 years. The record is based on branched GDGTs in Lake Gahai, and is compared with other temperature records from the TP to evaluate spatial patterns in the temperature evolution. They find that there are many distinct temperature patterns on the TP and suggest that people need to study their proxy well before interpreting the record.

The paper is clear and well written., but I have some recommendations that will hopefully take the paper one step further.

General recommendations:

The authors currently only discuss the trends in their record, but they do not discuss absolute values, despite the fact that they spend an entire section of the discussion on considering different calibrations. I would like to read more on how the brGDGT-based temperatures relate to the other temperature records. And see the absolute changes in the record interpreted. For example, is the 10 C temperature difference in the record realistic? And why (not)? And if true, what are the implications for our understanding of the climate during the last deglaciation at the TP?

**Response:** Thanks for your meaningful comments. Our study reconstructed the mean temperature of Months Above Freezing in Gahai lake using a new Bayesian calibration (Martínez-Sosa et al., 2021). The results of this reconstruction indicate temperatures higher than the annual average temperature and lower than the average temperature of summer months (June to August). The average annual temperature in the Gahai region is 1.2°C, and the average temperature during the summer months is 9.9°C. The temperature we reconstructed using surface sediments is 8.8°C, which aligns with the mentioned conditions.

As for the absolute temperature changes since 15,000 yr, although some influential studies indicate a warming of approximately 6.1-7°C from the deglaciation onset to preindustrial times (Tierney et al., 2020; Osman et al., 2021). However, these results are based on global mean sea surface temperatures. Our reconstructed temperature range is about 10°C, considering the remarkable 'elevation-dependent warming' observed in high-altitude regions compared to low-altitude areas (Mountain Research Initiative EDW Working Group, 2015). Thus, this range could be accurate. Nevertheless, we do not rule out the possibility that our temperature reconstruction may exhibit an overestimation. This is a known issue in temperature reconstruction using biomarkers. Aside from potential uncertainties associated with the biomarkers themselves, calibrations may also significantly influence the observed amplitude. We examined temperature variations reconstructed using different calibrations (Fig. S3), with the smallest range being 6°C and the largest being 12°C. Undoubtedly, further efforts are needed to constrain the inherent uncertainties related to biomarker-based temperature reconstructions.

Our preliminary idea is that using regional or global transfer equations for temperature reconstruction may potentially lead to similar issues. Instead, conducting site-specific calibration for temperature reconstruction may help reduce the amplitude of temperature variability.


"The brGDGTs-based temperature record from Gahai confirms the occurrence of a climate optimum in the mid-Holocene on the northeast Tibetan Plateau, which is consistent with several other pollen and pollen-reconstructed temperature records from the fringe areas of the Asian summer monsoon (Fig. 6), suggesting that it is a reliable representation of Holocene temperature changes in this region."

Other comments:

Comment: Introduction: The rationale on which exact temperature different proxies represent is not well explained. For example, the chironomid record from Tiancai lake is presented as July temperature. How valid is this? The growing season of chironomids may change over time, especially over glacial/interglacial transitions. Is it always July? The same is true for brGDGTs and pollen. Do they present annual or seasonal (and in that case: ice-free, or summer, or growing season) temperatures? How can we tell, and can we assume that seasonality is constant over time? This is important because climate modelers will want to use these records and it is our responsibility to make sure that they use the right temperatures to warrant low uncertainty on model projections of future climate. For example, the set of aims listed at the end of the introduction raises the question if ice-free season temperatures equal a warm-bias? And why compare the brGDGT record with July temperatures? What is the reason behind that?

**Response:** Thank you for your suggestion. The issue you mentioned is of paramount importance and remains an ongoing challenge yet to be resolved. The method of transferring contemporary knowledge and principles to the study of the past, known as the 'present is the key to the past', stands as a foundational and paramount methodology within the realm of Earth sciences. This approach involves deducing the conditions, processes, and characteristics of geological events in ancient times based on contemporary understanding and principles.

While the application of current knowledge to the study of the past is viable to a certain extent, it bears the potential for inadvertent deviations from factual precision. Consequently, addressing this issue in the context of the study of lacustrine sediments from the Holocene epoch requires an initial assessment of sedimentary environment stability based on factors such as existing sedimentation rates, chronology, lithological variations, and more. This evaluation helps prevent significant sedimentary discontinuities or other impactful events that could affect the reliability of proxy indicators.

The mentioned study on chironomid record from Tiancai lake, as referenced in Zhang et al., 2017a, explicitly demonstrates that chironomid respond to July temperatures. The chironomid-based transfer function, developed from the region of southwestern China based on a 100-lake calibration training set (Zhang et al., 2017b) was employed to translate the chironomid assemblage data from Tiancai lake into a reconstruction of mean July temperatures. This transfer function, constructed via a weighted averaging partial least squares (WA-PLS) regression in C2 (Juggins, 2005), spans a range of mean July temperature from 4.2 to 20.8 °C, whereas Tiancai lake currently exhibits a modern mean July temperature value of 8.3 °C. This modern process has been applied to the Tiancai lake sediment core, thus enabling the reconstruction of July temperatures over the past 9,000 years. The original authors consider this reconstruction to be reliable.

Similarly, through the examination of modern processes involving global lake and soil, it has been concluded that both brGDGTs and pollen respond to temperature changes. Consequently, these indicators have been employed extensively for temperature reconstructions in Quaternary paleoclimate research, embracing the 'present is the key to the past' principle. However, questions regarding the consistent existence of seasonality remain unresolved. This underscores one of the limitations of paleoclimate reconstruction work, motivating a clear understanding of the indicative significance, sensitivity, and representativeness of relevant proxy indicators. This will help mitigate uncertainties and provide more accurate records for paleoclimate simulation studies.

Finally, temperatures during the non-freezing seasons tend to exceed annual averages yet remain lower than summer temperatures. We are inclined to attribute this to warm-season bias, as opposed to a warm bias. As for why brGDGT-based temperature reconstructions are compared to July temperatures, one prevailing hypothesis addressing the "Holocene temperature conundrum" highlights the possible role of seasonal discrepancies in proxy indicators. These discrepancies could arise from varying climatic implications intrinsic to the proxy indicators themselves. Therefore, a

comprehensive understanding of the climatic significance, sensitivity, and representativeness of proxy indicators is pivotal. Undertaking multi-proxy temperature reconstructions within the same study area could shed light on potential seasonal deviations in different temperature series and contribute to unraveling the "Holocene temperature conundrum". Presently, the bulk of studies rely on single temperature proxies. In contrast, we have utilized two distinct temperature proxies, brGDGTs and pollen, from the same borehole to reconstruct Holocene temperature, thus vividly illustrating the existence of seasonal biases in different indicators.


Comment: BrGDGTs: it should be explicitly stated in the introduction that brGDGTs are produced in lakes. The introduction only mentions that certain factors can influence their distribution, but this fact must be linked to their producers that are sensitive to e.g., redox conditions (Weber et al., 2018 PNAS), O2 content and/or mixing regime (van Bree et al., 2020 Biogeosciences, Wu et al., 2021 Chem Geol.), and, consequently, also to lake depth due to the different niches that the producers of different brGDGTs occupy. It is the structural offset between brGDGT distributions in lakes and in soils that has led to the lake-specific calibrations (see Tierney and Russell, 2009, Sinninghe Damsté et al., 2009, or Tierney et al., 2010 for the early works).

**Response:** Thanks for your meaningful comments. In the introduction section, we underscored the complexity of the sources of brGDGTs within lakes. Moreover, an increasing body of research indicates that brGDGTs in lakes are primarily of autochthonous origin. In response to reviewer comments, we have clarified that the factors influencing the distribution of brGDGTs are closely tied to their producers. Subsequently, we elaborated on the potential sources of brGDGTs in this study and the environmental conditions under which their producers thrive. The specific revisions made in this section are outlined below:

Lines 128-139: "However, the factors that impact the distribution of brGDGTs in lakes are intricate and multidimensional. Notably, the sources of brGDGTs within lakes are intricate, involving contributions from soil as well as autochthonous lake

processes. Moreover, an expanding body of research underscores a significant prevalence of autochthonous brGDGTs in lakes (Tierney and Russell, 2009; Tierney et al., 2010; Weber et al., 2015; Wang et al., 2021b). Furthermore, the origins of brGDGT producers remain uncertain and could be influenced by various factors, including lake salinity (Wang et al., 2021b), redox conditions (Weber et al., 2018), oxygen content and/or mixing patterns (Buckles et al., 2014; van Bree et al., 2020; Wu et al., 2021). Additionally, even lake depth plays a role due to distinct ecological niches (Woltering et al., 2012), thereby contributing to the intricate interplay that shapes the distribution of brGDGTs within lakes."


As detailed in the section about study site, Gahai lake is located on the eastern edge of the Tibetan Plateau, with consistently low temperatures throughout the year. The maximum depth of the lake does not exceed two meters. From October to Aprial in the next year, the lake is predominantly covered by ice. Consequently, the sediment cores likely record the average temperatures during the ice-free months. Furthermore, Gahai lake is presently an open lake, with its waters flowing northwestward through the Tao River and eventually into the Yellow River. Thus, we have reason to believe that the lake's water level has not exceeded two meters since the Holocene. Given the shallowness of the lake in such a cold region, we posit that brGDGTs in the lake sediment are more likely to grow during the warmer seasons, while the complete freezing of the lake during ice-cover periods restricts the growth of brGDGT producers.

Comment: How can brGDGTs represent both annual mean air temperature and a warm-biased temperature (L35-36 in the abstract)?

**Response:** Thank you for your reminder. Here, we are referring to instances where brGDGTs are utilized for various interpretations in different studies. For example, in some articles, brGDGTs are considered to reflect annual average temperatures, while in others, they are believed to indicate warm-biased temperatures. This might have caused ambiguity in our original statement. Thus, we have rephrased the sentence for clarity. "The results demonstrate that brGDGTs have been employed to reconstruct various temperatures in different studies, including annual average temperature and warm-biased temperature."
 Please see lines 35-37.

Comment: L169: add number of soils sampled and indicate the depth interval, and also plot them on the triplot (Fig. 4). Actually, it is quite clear that such a triplot does not really separate brGDGTs in lake sediments from soils on a global scale – it seems best suited to identify in situ production in the marine coastal zone (Sinninghe Damsté, 2016). However, lakes tend to have more hexamethylated brGDGTs (in particular IIIa), so directly comparing the relative abundances of brGDGTs and soils with a bar-plot could be more informative (see e.g., Fig. 11 in Martinez-Sosa et al., 2021). It can also help to do Principle Component Analysis (PCA) on the lake sediment record, and then passively add the soils to see where they plot. The sample distribution in the PCA can further help to identify the main changes in brGDGT distributions through time and possibly connect this to likely environmental drivers (and finally temperature).

**Response:** Thanks for your suggestions. Four catchment soil samples were collected from around the lake. As per your suggestion, we have explicitly mentioned this number (four catchment soils) in the manuscript and plot them on the triplot (see Fig. 1). In addition, we also compared the relative abundance of brGDGTs in soils and core sediment samples with a bar-plot in Fig 2 here, which is also presented in Supplementary Fig. S2.

[Figure]

Fig. 1 Comparison of the fractional abundances of tetramethylated, pentamethylated, and hexamethylated bGDGTs in sediment core samples and catchment soils from Gahai with lake surface sediments from the Tibetan Plateau (Günther et al., 2014; Wang et al., 2016), East Africa (Russell et al., 2018), and worldwide (Martínez-Sosa et al., 2021).

[Figure]

Fig. 2 Mean fractional abundances and standard deviations of brGDGTs from downcore sediments and catchment soil samples in Gahai lake.

Comment: The methods can do with a bit more information on the lithology and the age model. In the Discussion the oldest 5000 years are suddenly discarded (L308), as these sediments could represent a slump, but no evidence is provided.

**Response:** We are very grateful to you for your meaningful comments. We have included this part in the text, please see lines 187-213.

"The chronology of the upper 20 cm of the sediment core is based on measurements of $^{210}Pb$ and $^{137}Cs$, at a 1-cm interval. The chronology for the deeper part

of the core is provided by accelerator mass spectrometry (AMS) [14]C measurements of 13 bulk sediment samples, which were conducted by Beta Analytic Inc. (Miami, USA) (Fig. 2) (Wang et al., 2022).

The [210]Pb age model was constructed using the constant rate of supply (CRS) model and the [137]Cs peak was used as supplement (Appleby, 2002). The calculated age of [210]Pb using CRS model aligned well with the [137]Cs peak at 6 cm. Overall, the CRS model was deemed suitable for determining the age of Gahai lake.

Reservoir age, as highlighted by Hou et al. (2012), is a crucial factor affecting the age determination of lake sediment cores on the TP. Therefore, it was necessary to establish the reservoir age of Gahai lake before undertaking paleoclimate reconstruction. The linear extrapolation relationship between the [14]C ages and depth to the sediment-water interface is often used to estimate the reservoir age. The [14]C age of 13 samples exhibits a good linear relationship with sediments depth in Gahai lake. Extrapolation of this 13 [14]C ages down to the depth of 6 cm yielded a [14]C age of 461 yr BP, while the reliable [210]Pb age at 6 cm is -27 yr BP. Consequently, the difference between the two ages, which amounts to 488 yr, was taken as the reservoir age. Additionally, it's worth noting that independent estimations of the [14]C calibration age and [210]Pb age around 10 cm in Gahai lake was obtained, resulting in values of 497 yr BP and 18 yr BP, respectively. The difference of 479 yr between these two ages can also be considered as the reservoir age. These two methods of estimating reservoir age of Gahai lake show very close, which are mutually supportive. So, the average of 483 yr was adopted as the reservoir age. All original [14]C dates were corrected by subtracting the reservoir age (483 yr) and calibrating them to calendar ages using Calib 8.1. The age-depth model (Fig. 2) was constructed using the Bacon program with the [14]C ages and [210]Pb ages (Blaauw and Andres Christen, 2011) and was reported by Wang et al. (2022)."

Regarding the lithology, especially the sudden exclusion of the oldest 5000 years, we believe that the corresponding time interval (from 191 cm) may have undergone rapid sedimentation or alternatively slumping. There are several reasons for this:

1. We can observe that the ages around 191 cm, 229 cm and 279 cm are relatively close (15070 a, 14870 a, 15500 a, respectively), which suggests the possibility of rapid sedimentation.
2. By examining the grain size distribution, we can notice significant fluctuations in the silt fraction (4-63 μm) starting from 191 cm (Fig. 3). The silt fraction in Gahai is driven by the medium silt (16-32 μm) fraction, while the fine and coarse silt fractions remain almost unchanged during the Holocene, hence the fine, medium, and coarse silts are combined into the total silt fraction (4-63 μm) for discussion (Wang et al., 2022). This could indicate the occurrence of specific events.

[Figure]

Fig. 3 Silt fraction (4-63 μm) distribution.

3. During this stage, the concentration of brGDGTsis notably low, which hinders our ability to conduct thorough analysis. Similarly, the pollen concentration during this time period is also quite low, and the data are insufficient for statistical analyses (Wang et al., 2022).

Therefore, we only present the research results from the past 15,000 years.


Comment: Interpretation of the temperature record by vegetation change in the area, and thus heat capacity (L333): Note that this process is mostly valid for soils, for which this has been described, but that the vast majority (if not all) of the brGDGTs in the lake record will be produced in situ. Assuming that the lake has always contained water during the studied interval, the heat capacity would not have changed.

**Response:** Thank you very much for your suggestion, and we have removed this statement.

Remediation effects: Since this anthropogenic spike only represents a few datapoints I would be careful not to overinterpret the data. Especially stating that remediation effects have an effect and that brGDGTs now record temperature again based on only the surface sediment would be overstating it for me. Please tone down and substantially reduce the discussion on this part of the record.

**Response:** Thank you for your reminder. In accordance with your suggestion, we have reduced the discussion in this section.

Minor comments:

**Comment: Check the order of references. By first name, then by year.**

**Response:** Thank you for your suggestions, we have made individual revisions throughout the manuscript.

**Comment: Often use of 'significant' without providing p-value.**

**Response:** Thank you for your reminder, we have thoroughly checked and made corresponding revisions to each of them.

**Comment: Damsté et al should be Sinninghe Damsté et al.**

**Response:** Thank you very much for your reminder. We have conducted a thorough review and made all the necessary modifications.

**Comment: Provide information on the detection limit used for GDGT quantification.**

**Response:** Thank you for your suggestion. During the sample testing process, the detection limit for GDGTs testing is 0.0004 ppm.

From: Dr. Juzhi Hou

Institute of Tibetan Plateau Research

Chinese Academy of Sciences

11/24/2023

To: Dr. Qiuzhen Yin

Editor

Climate of the Past

Dear Dr. Qiuzhen Yin,

On behalf of the co-authors, we are very grateful to you for giving us an opportunity to revise our manuscript. We really appreciate your positive and constructive comments together with suggestions on our manuscript entitled 'BrGDGTs-based seasonal paleotemperature reconstruction for the last 15,000 years from a shallow lake on the eastern Tibetan Plateau' (MS No.: cp-2023-32). We have therefore studied reviewer' comments carefully and tried our best to revise our manuscript accordingly. Notably, the changes are highlighted in track-changes manuscript. Please see below for a point-by-point response to the reviewers' comments and concerns.

**Editor and Reviewer comments:**
**Reviewer#2:**
Overall, I am satisfied with the replies of the authors to my questions and comments and would like to thank the authors for their efforts. However, there are a few issues remaining that I recommend following up upon prior to accepting this manuscript for publication.
- L35: replace 'The results demonstrate…' by 'In these studies, brGDGTs have been interpreted to reflect either mean annual air temperature or growing season temperature. In both cases, brGDGTs reflect a gradual warming trend….'
**Response:** Thank you for your suggestion. we have rephrased this.

This omits the use of 'warm bias' in your text, adding to my earlier comment on

referring to mean air temperatures for months above freezing (MAF) as reflecting a 'warm bias'. The use of this term is very confusing as it raises the question where the bias is compared to. I'm guessing the authors mean MAT, but if MAF is reconstructed, this comparison is not valid. Thus, please refer to MAF, as the proxy actually reconstructs and avoid comparing apples with oranges.

**Response:** Thanks for your meaningful comments. We have changed 'warm-biased temperature' into 'the mean air temperatures for months above freezing (MAF)' throughout the text.

- L136: The 'moreover' should be a 'however', as this sentence contradicts part of what you are saying in the previous sentence (brGDGTs in lakes have mixed sources vs brGDGTs in lakes are produced in situ).

**Response:** Thanks for your reminder, we have changed "moreover" into "however".

- L266 + L272: change abundance to abundant

**Response:** Thanks for the suggestion, we have corrected it.

- Fig. 3. Which compound does IIa'' refer to? The molecular structure is not given in Fig. S2. How would it look like?

**Response:** Thanks for your reminder. The compound IIa" in Fig. 3 should be IIa''' and we have corrected it. The compound IIa''' was only found in Gahai catchment soil samples and is shown in the revised Fig. S2.

[Figure]

Fig. 3 Representative high-performance liquid chromatography/atmospheric pressure chemical ionization-mass spectrometry (HPLC/APCIMS) chromatograms of brGDGTs from (a) surface sediments from Gahai lake, and (b) soils in the catchment of Gahai lake.

[Figure]

Fig. S2 Mean fractional abundances and standard deviations of brGDGTs in the downcore sediments and 19 catchment soil samples at Gahai lake.

- L319: the unknown producers of brGDGTs and their response to changes in autotrophic biomass production has already been mentioned in L313. Remove the repetition.

**Response:** Thanks for your meaningful comments. We have deleted this sentence.

- L342-377: in my opinion, this part of the discussion is redundant as you have just

made the argument that the calibration of Martinez Sosa et al should be used to reconstruct MAF. Partially repeating the comment in my previous review:

I follow (and agree with) the rationale of the authors to use the Martinez-Sosa calibration. Note, however, that this study is based on the fact that brGDGTs in lakes are in situ produced (i.e., have an autochthonous source), and that brGDGT distributions in lakes and soils are substantially different (see the discussion in their section 4.1). Hence, this makes the use of additional calibrations redundant, in particular considering that the fact that most brGDGTs in Gahai Lake have an autochthonous source, is used as the main motivation. Since most of these calibrations are based on (a variation of) the MBT'5me, this exercise mainly just changes the absolute temperature values rather than doing anything else (such as revealing new insights).

Thus, I still suggest the authors to make a clear, motivated decision on the choice of calibration (you can even mention that other calibrations exist but are based on the same principles, just using a different dataset, and will thus generate a record with the same trends just different absolute temperatures), and then interpret just that record. Leave all discussion about the other calibrations out to keep the discussion focused on the interpretation of the trends and timing of changes in the temperature record.

**Response:** We thank Reviewer for the meaningful comments and suggestions. We have rephrased this part in the revision as follows:

'Using calibration of Martínez-Sosa's et al. (2021), we reconstructed the surface sediment temperature of Gahai lake, resulting in a temperature estimate of 9.4°C. This reconstructed temperature closely matches the ice-free season temperature recorded by meteorological stations in the Gahai region (8.8°C for May to September). Furthermore, considering the significant contribution of autochthonous brGDGTs in Gahai lake, we also attempted to reconstruct the Holocene paleotemperature record using previously published lake-specific brGDGTs-temperature calibrations (e.g., Günther et al., 2014; Martínez-Sosa et al., 2021; Russell et al., 2018; Sun et al., 2011; Wang et al., 2016). As depicted in Fig. S3, most of these calibrations exhibit qualitatively similar temperature change patterns when applied to the sediment core from Gahai Lake. This similarity

arises from their shared same principles, just utilizing distinct datasets, resulting in records that display analogous trends but vary in absolute temperatures.'

- L459: 'this is a known issue in temperature reconstruction using biomarkers'. I am not sure where this statement comes from and why this is a known issue. There are many different biomarkers that are used for temperature reconstruction in both terrestrial and marine realms, but I have never noticed this issue, which should be quite prominent if true.

**Response:** Thanks for your meaningful comments. We have deleted this sentence.

From: Dr. Juzhi Hou
Institute of Tibetan Plateau Research
Chinese Academy of Sciences

Dec 19, 2023

To: Dr. Qiuzhen Yin
Editor
Climate of the Past

Dear Dr. Yin,

Thank you very much for your comments and suggestion on our manuscript entitled "BrGDGTs-based seasonal paleotemperature reconstruction for the last 15,000 years from a shallow lake on the eastern Tibetan Plateau" (No. cp-2023-32). We have carefully addressed your comments point-to-point below and highlighted in the text. We wish the revised manuscript meets the publication criteria for Climate of the Past.

**Comments:**
Line 36-38: This sentence is not clear. "a warming trend during the Holocene" is told at the beginning, but "a cooling trend" is said at the end.

**Response:** Thank you for your suggestion. We have rephrased the sentence, "The existing/available temperature records show complicated patterns of variation, some with general warming trends throughout the Holocene, some with cooling trends, while some with warm middle Holocene".

The abstract is too long and sometimes confusing. I suggest to make it more concise.

**Response:** Thank you for your suggestion. We have shorten the abstract, please see the revised text.

Line 430-431: Apparently this is not what Fig.6 tells. this sentence needs to be modified.

**Response:** We appreciate the comment and we have deleted this sentence, which isn't suitable here. Additionally, we have merged the next paragraph with the previous one.

Figure 7: What are the red and blue curves? How is this Jun-Sep insolation calculated and what is the unit? I tried different ways to make the calculation, but I could not get similar result as the authors. What is the Jul insolation and its unit? monthly mean, mid-month daily insolation, total insolation? Why do you need to show both insolation curves?

**Response:** Thank you very much for your advice and assistance. We have clarified the confusion in Fig. 7. The red curve represents the cumulative insolation from June to September in $W/m^2$, and the navy blue curve shows the mean insolation during July in $W/m^2$. Both are shown for comparison with temperature reconstruction in ice-free season (brGDGTs in this study) and in July (pollen in Wang et al., (2022)) at Lake Gahai. The difference in the two temperature records may result from the lags between the insolation in June to September and in July.

The previous solar radiation data was obtained from https://vo.imcce.fr/insola/earth/online/earth/online/index.php. The 34°N July mean insolation and June-September mean insolation calculated in Fig.7 are obtained by dividing the total solar radiation for July 1st to July 31st and June 1st to September 30th, respectively, by the length of these two time periods. The total solar radiation is calculated using the Berger et al. (2010) elliptic integral method.

We have already labeled this in the caption of Fig. 7f, named "Mean insolation during July (W/m2) (navy blue curve) and mean insolation during ice-free months (W/m2) at 34 °N (red curve)". Relevant references have been cited.

I draw the authors' attention that Copernicus Publications request depositing data that correspond to journal articles in reliable (public) data repositories. Please see the data policy at https://www.climate-of-the-past.net/policies/data_policy.html.

**Response:** Thanks for your reminder. We have prepared the data and will submit it as requested.